# SVBench: A Benchmark with Temporal Multi-Turn Dialogues for Streaming Video Understanding

**Zhenyu Yang[1,2,3]\* Yuhang Hu[4], Zemin Du[5], Dizhan Xue[1,2], Shengsheng Qian[1,2]†, Jiahong Wu[3], Fan Yang[3], Weiming Dong[1,2], Changsheng Xu[1,2,6]**

[1]Institute of Automation, Chinese Academy of Sciences, [2]University of Chinese Academy of Sciences, [3]Kuaishou Technology, [4]Zhengzhou University, [5]ShanghaiTech University, [6]Peng Cheng Laboratory
`yangzhenyu2022@ia.ac.cn`
`shengsheng.qian@nlpr.ia.ac.cn`

## ABSTRACT

Despite the significant advancements of Large Vision-Language Models (LVLMs) on established benchmarks, there remains a notable gap in suitable evaluation regarding their applicability in the emerging domain of long-context streaming video understanding. Current benchmarks for video understanding typically emphasize isolated single-instance text inputs and fail to evaluate the capacity to sustain temporal reasoning throughout the entire duration of video streams. To address these limitations, we introduce SVBench, a pioneering benchmark with temporal multi-turn question-answering chains specifically designed to thoroughly assess the capabilities of streaming video understanding of current LVLMs. We design a semi-automated annotation pipeline to obtain 49,979 Question-Answer (QA) pairs of 1,353 streaming videos, which includes generating QA chains that represent a series of consecutive multi-turn dialogues over video segments and constructing temporal linkages between successive QA chains. Our experimental results, obtained from 14 models in dialogue and streaming evaluations, reveal that while the closed-source GPT-4o outperforms others, most open-source LVLMs struggle with long-context streaming video understanding. We also construct a StreamingChat model, which significantly outperforms open-source LVLMs on our SVBench and achieves comparable performance on diverse vision-language benchmarks. We expect SVBench to advance the research of streaming video understanding by providing a comprehensive and in-depth analysis of current LVLMs. Our benchmark and model can be accessed at `https://yzy-bupt.github.io/SVBench`.

## 1 INTRODUCTION

In recent years, the rapid advancements in Large Language Models (LLMs) Touvron et al. (2023); Achiam et al. (2023) and visual processors Radford et al. (2021); Dosovitskiy et al. (2020) have significantly enhanced the performance of Large Vision-Language Models (LVLMs) Zhu et al. (2023); Ataallah et al. (2024); Maaz et al. (2023). These powerful models have been instrumental in pushing the boundaries of artificial intelligence, showcasing exceptional progress in domains such as visual reasoning and dialogue, particularly in the field of video understanding.

Furthermore, there is a growing interest in applying these advancements to the emerging field of streaming video understanding Qian et al. (2024); Chen et al. (2024a). In conventional video understanding tasks, models are given the entire video and can leverage past, current, and future content to comprehend a specific video segment. In real-world scenarios, such as live streaming and security monitoring, video content is continually streamed, necessitating that dialogues should update

---

\*Work done during an internship at Kuaishou Technology.
†Corresponding author.

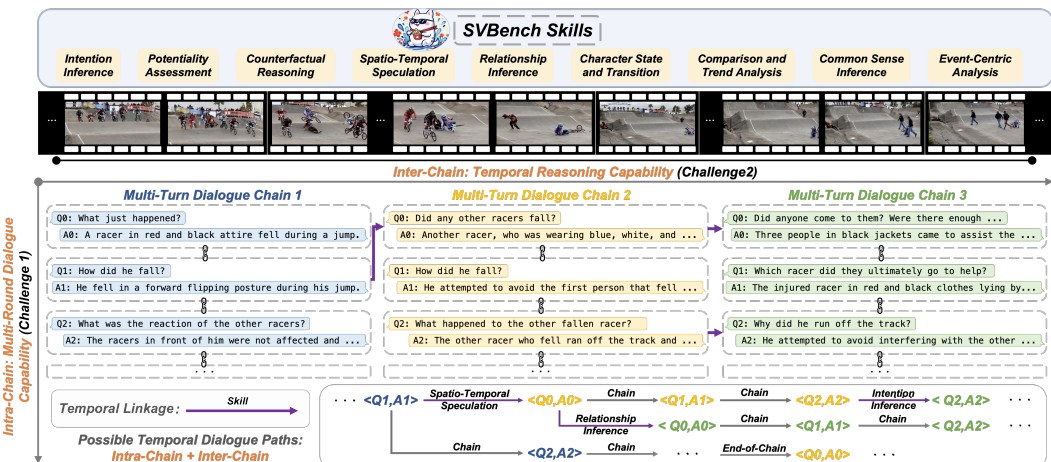

Figure 1: **Illustration of temporal multi-turn dialogues.** A temporal dialogue path represents a conversation within a video progressing over time. Our SVBench evaluates the capabilities of LVLMs in long-context streaming video understanding by constructing temporal dialogue paths to assess 9 critical skills.

concurrently with the temporal flow of the video without knowing the future information. Recently, streaming videos have become increasingly popular with diverse sources, including online video platforms Zellers et al. (2022), live streaming services Gao et al. (2023), and wearable camera footage Grauman et al. (2022). Therefore, researchers have begun to study LVLMs with the ability to interpret and interact with streaming content Qian et al. (2024); Chen et al. (2024a). Given these research advancements, it becomes imperative to establish a comprehensive evaluation benchmark specifically tailored to assess the progress in streaming video understanding achieved by LVLMs.

Existing video benchmarks primarily focus on single-turn text input and cannot measure the ability to conduct temporally sequential reasoning, which falls short of capturing the complexity of streaming video understanding. For instance, several question-answering benchmarks Yu et al. (2019); Zhang et al. (2023); Xiao et al. (2021) typically comprise disjointed QA pairs tied to individual video clips, ignoring the continuous and dynamic nature of video streams. However, in real-world scenarios, users usually ask multiple questions during the entire duration of a video stream, with potential relevance among questions and video clips. Such complex human-computer interaction necessitates mastering fundamental skills, including engaging in multi-turn dialogues and comprehending extensive contextual histories to foster coherent and contextually relevant conversations. Although some current benchmarks partially assess essential interaction abilities, they are limited by relying on either multiple static images Feng et al. (2022) or short clips Zhao et al. (2018) to simulate multi-turn dialogues, rather than using long streaming videos. Therefore, appropriate benchmarks are required to evaluate the ability of streaming video understanding. To tackle these problems, we propose the first benchmark for streaming video understanding, named **S**treaming **V**ideo understanding **Bench**mark (**SVBench**), which aims at comprehensively evaluating the temporal multi-turn dialogue capabilities of LVLMs for streaming videos.

First, we introduce a novel and challenging task named temporal multi-turn dialogue for streaming videos. We define a QA chain to represent a series of consecutive multi-turn dialogues over a video segment. Subsequently, we define *Temporal Linkages* between successive QA chains for video segments, which can be established based on the common people, events, objects, etc. LVLMs should understand the current video segment, dialogue, as well as historical video segments and dialogues to answer the current question. For example, as shown in Figure 1, to answer the question *"Did any other racers fall?"*, an LVLM should recall the previously called racer in historical video segments and dialogues. Our proposed task aims to comprehensively evaluate the capability to leverage historical content and conduct multi-turn dialogue throughout a real-time streaming video.

Second, we construct a large-scale dataset with temporal multi-turn question-answering chains for the proposed streaming video understanding task. We compare our dataset with existing video datasets in Table 1. Our dataset comprises 1,353 diverse videos from 6 streaming platforms, each undergoing thorough filtering and meticulous selection. Coupled with streaming videos are anno-

Table 1: The comparison of different datasets. **Avg. Q/V**: the average number of QA pairs per video. **Open-Domain**: whether the video sources are diverse. **Long**: whether the average video length is greater than 2 minutes. **Dialogue**: whether there are contextual connections between QA pairs. **Streaming**: whether the QA pairs can be tested in sync with the video over time.

| Dataset | #QAs | Avg. Q/V | Long | Open-Domain | Dialogue | Streaming | Annotation |
|---|---|---|---|---|---|---|---|
| EgoSchema Mangalam et al. (2024) | 5,063 | 1.00 | ✔ | ✗ | ✗ | ✗ | Auto&Manual |
| ActivityNet-QA Yu et al. (2019) | 800 | 1.00 | ✗ | ✗ | ✗ | ✗ | Manual |
| MVBench Li et al. (2024) | 4,000 | 1.10 | ✗ | ✔ | ✗ | ✗ | Auto |
| How2QA Li et al. (2020) | 44,007 | 2.00 | ✗ | ✔ | ✗ | ✗ | Manual |
| Perception Test Patraucean et al. (2024) | 44,000 | 3.79 | ✗ | ✗ | ✗ | ✗ | Auto&Manual |
| Social-IQ Zadeh et al. (2019) | 7,500 | 6.00 | ✗ | ✗ | ✗ | ✗ | Auto&Manual |
| MSVD-QA Xu et al. (2017) | 13,157 | 6.68 | ✗ | ✔ | ✗ | ✗ | Auto |
| TVQA Lei et al. (2018) | 152,545 | 7.00 | ✗ | ✗ | ✗ | ✗ | Manual |
| NExT-QA Xiao et al. (2021) | 52,044 | 9.57 | ✗ | ✔ | ✗ | ✗ | Manual |
| MovieChat Song et al. (2024) | 13,000 | 13.00 | ✔ | ✗ | ✗ | ✗ | Manual |
| LVBench Wang et al. (2024b) | 1,549 | 15.04 | ✔ | ✔ | ✗ | ✗ | Manual |
| TGIF-QA Jang et al. (2017) | 165,165 | 17.25 | ✗ | ✔ | ✗ | ✗ | Auto&Manual |
| MSRVTT-QA Xu et al. (2017) | 72,821 | 24.35 | ✗ | ✔ | ✗ | ✗ | Auto |
| **SVBench (Ours)** | **49,979** | **36.94** | ✔ | ✔ | ✔ | ✔ | **Auto&Manual** |

tations comprising 49,979 QA pairs, where, on average, each video contains 36.94 pairs, which is the highest number among known video datasets. Moreover, we build temporal dialogue paths that occur in sync with a video progressing over time, as shown in Figure 1, designed specifically to assess the capability to reason effectively through time.

Third, we conduct extensive evaluations, in terms of dialogue and streaming performance, of various prevalent LVLMs on our SVBench. Our results provide the first overview insight into the streaming video understanding capability of existing LVLMs. Surprisingly, these state-of-the-art LVLMs are far from satisfactory, in terms of streaming video understanding. These results motivate us to develop a stronger LVLM, namely **StreamingChat**, which significantly improves the overall dialogue evaluation score by 9.41% and the streaming evaluation score by 3.30% on our SVBench, compared to the top-performing open-source LVLMs, while achieving comparable performance on conventional image and video benchmarks. Our benchmark and model will be publicly available, in order to catalyze the progress in streaming video understanding.

## 2 RELATED WORK

### 2.1 LARGE VISION-LANGUAGE MODELS FOR VIDEO

Recent advancements in Large Language Models (LLMs) have paved the way for a significant research focus on Large Vision-Language Models (LVLMs) aimed at improving multimodal understanding, such as multimedia retrieval Yang et al. (2024b;a), particularly in video content. Currently, in addition to the popular closed-source LVLMs such as GPT-4V Yang et al. (2023), GPT-4o Achiam et al. (2023), and Gemini 1.5 Pro Reid et al. (2024), an increasing number of open-source LVLMs, including Video-ChatGPT Maaz et al. (2023), VideoLLaMA2 Cheng et al. (2024), and VILA Lin et al. (2024), also have demonstrated the impressive capability in video understanding tasks. Advanced human-computer interaction in everyday life requires the ability to engage in multi-turn dialogues and understand extensive contextual histories to maintain coherent and contextually appropriate conversations Qian et al. (2024); Chen et al. (2024a). However, these LVLMs are still not fully adept at handling the intricacies of streaming videos and do not completely grasp the complexities of real-world contexts. To rigorously evaluate the capabilities of these models, we propose SVBench to measure the performance of LVLMs in video-related tasks that imitate the complexity of real-world interactions.

### 2.2 VIDEO UNDERSTANDING BENCHMARKS

In recent years, the exponential growth of video data has elevated video understanding to a crucial area within computer vision. To rigorously assess the performance of LVLMs, researchers have

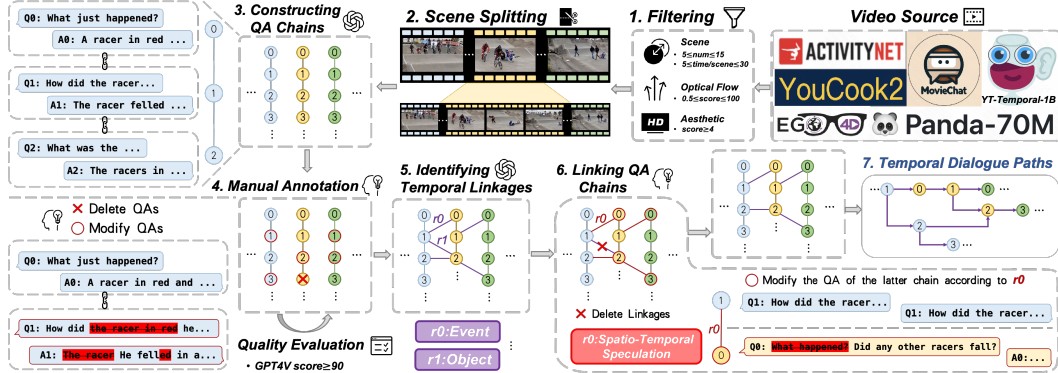

Figure 2: **Overview of the proposed SVBench framework:** (1) Filtering raw videos from diverse streaming sources; (2) Detecting scenes and splitting videos accordingly; (3) Constructing QA chains for dialogues within videos; (4) Performing manual annotation and quality assessment; (5) Identifying temporal linkages between QA chains; (6) Connecting QA chains to facilitate temporal reasoning; (7) Building temporal dialogue paths for evaluating LVLMs.

introduced a range of standardized benchmarks. These benchmarks provide not only comparative evaluation criteria for models but also catalyze advancements in video understanding models. The recent benchmarks, such as TGIF-QA Jang et al. (2017), MSVD-QA Xu et al. (2017), and MVBench Li et al. (2024), primarily comprise relatively brief videos capturing single events, thereby overlooking the temporal dependencies inherent in longer videos. To address the intricacies of long video comprehension, benchmarks using longer videos like movies have been developed. For instance, LLaMA-Vid Li et al. (2023), based on MovieNet Huang et al. (2020), has developed a movie QA dataset to identify character relationships. Similarly, MovieChat Su et al. (2020) employs a diverse set of videos and avoids specific character names or plot details within its questions. However, these benchmarks often fall short in addressing the intricate challenges posed by streaming videos, which encompass extended temporal contexts and dynamic scene variations. Therefore, we establish SVBench, a novel and comprehensive benchmark that aims to bridge this gap by offering an elaborate evaluation framework for long-context streaming video understanding.

# 3 DATASET

We develop a comprehensive data collection pipeline to construct a high-quality streaming video dataset, tailored for annotating temporal multi-turn dialogues. We split the SVBench dataset into a training set and an evaluation set, ensuring that videos and their corresponding QA pairs appear in only one split. This results in 42,605 QA pairs for training and 7,374 QA pairs for evaluation, with 1,153 videos and 200 videos in each set, respectively.

## 3.1 DATA FILTERING AND SCENE SPLITTING

We source 12,989 raw video data from a variety of publicly available datasets, including YT-Temporal-1B Zellers et al. (2022), YouCook2 Zhou et al. (2018), ActivityNet Caba Heilbron et al. (2015), MovieChat Song et al. (2024), Panda-70M Chen et al. (2024c), and Ego4D Grauman et al. (2022). These datasets offer a wide spectrum of video content, ensuring a rich diversity that is essential for streaming video understanding. We then filter high-quality videos of adequate lengths, high aesthetic scores, and appropriate optical flow scores. Subsequently, we employ PySceneDetect[1] to identify and enumerate scenes within filtered videos. These results are crucial in determining whether the video content exhibits adequate variation and complexity. Further filtering is conducted to retain only those videos containing 5 to 15 scenes, thereby excluding content that is either excessively monotonous or overly intricate. Moreover, only videos with an appropriate average scene duration are chosen, ensuring fluidity and rhythm. Finally, 1,353 videos are selected. Ultimately, we split each video into clips $V = \{s_i \mid 0 \le i < |V|\}$ based on timestamps, merging clips shorter

---

[1]https://github.com/Breakthrough/PySceneDetect

than 2 seconds with their adjacent ones. Notably, to prevent disjointedness and ensure continuity between scenes, we add an extra 0.5 seconds to both the beginning and end of each clip, resulting in a one-second overlap between consecutive clips. More details are included in Appendix A.

We propose a semi-automated annotation pipeline for streaming videos, as shown in Figure 2, including a multi-stage LLM-assisted generation process with several rounds of manual annotation. See Appendix H for detailed prompts. The annotation takes about 3 months and involves over 30 professional annotators.

### 3.1.1 CONSTRUCTING QA CHAINS FOR VIDEO DIALOGUES

We propose creating a multi-turn question-answering dataset on videos to evaluate the ability of LVLMs to conduct multi-turn dialogues and comprehend video-related contextual information. This ability is essential for the coherence and relevance of conversations in various contexts. Initially, we define a series of multi-turn question-and-answer interactions on video clips as QA chains. For every video clip $s_i \in V$, we build a QA chain $C_i = \{\langle Q_j^i, A_j^i \rangle \mid 0 \leq j < |C_i|\}$, where $Q_j^i$ and $A_j^i$ represent the $j$-th question and answer generated on the $i$-th video clip. To achieve this, we harness the video understanding capabilities of existing LVLMs (e.g. GPT-4o):

$$\mathcal{C} = \{C_i = \text{LVLM}(p_v, s_i) \mid 0 \leq i < M\}, \tag{1}$$

where $p_v$ represents the prompt that generates 5 to 6 consecutive rounds of questions and answers, and $M = |V|$ is the number of clips segmented from the video. However, due to the limitations of the video understanding and text generation capabilities of current LVLMs, we have to employ human annotators to manually augment, delete, and modify the QA pairs so that the QA in the chain is connected and aligned with the video. For instance, specific persons or objects mentioned in questions should be modified to utilize third-person pronouns (e.g. he/she/it/they).

### 3.1.2 IMPLEMENTING QA QUALITY EVALUATION

Due to the inconsistent quality of manually annotated QA chains, we devise a comprehensive evaluation mechanism to guarantee their high quality. We utilize GPT-4 to assess QA chain quality across 7 dimensions—accuracy, completeness, relevance, fluency, contextual comprehension, logical consistency, and temporal understanding, scoring each from 0 to 100. In addition to the 7 dimensions, the QA chain quality also has an overall score from 0 to 100, with a 90-point minimum for high standards. Any QA chain failing to meet this threshold must undergo manual revision again. This iterative process ensures the production of highly reliable and insightful QA chains, contributing to the utility of our SVBench.

### 3.1.3 IDENTIFYING TEMPORAL LINKAGES

Given that adjacent QA chains are derived from sequential video clips, they inherently contain overlapping entities such as objects, scenes, and events, as well as inherent temporal linkages. To effectively establish these temporal linkages, we initially employ LLMs (e.g. GPT-4) to search for and identify potential linkages between adjacent QA chains. For the junction of the $i$-th and the $(i+1)$-th QA chains, we construct a set of relations $R_i (0 \leq i < M - 1)$ as follows:

$$R_i = \{r_j \mid 0 \leq j < |R_i|\} = \text{LLM}(p_l, C_i, C_{i+1}), \tag{2}$$

where $p_l$ is the prompt that stimulates the generation of candidate relations and contextual categories between two given QA chains. Here, an individual relation $r_j$ is represented as a quintuple, structured as:

$$r_j = \langle Q_x^i, A_x^i, Q_y^{i+1}, A_y^{i+1}, Rc \rangle, \tag{3}$$

which suggests that a relation exists between the $x$-th QA pair within the QA chain of the $i$-th clip and the $y$-th QA pair within the subsequent QA chain, characterized by the relationship category $Rc$. Accordingly, we have delineated a range of relationship types to classify these relations, the outer ring as shown in Figure 3(b).

### 3.1.4 LINKING QA CHAINS FOR TEMPORAL REASONING

To evaluate the ability to reason through time, we need to establish temporal linkages between successive QA chains, which facilitates multi-turn QA interactions between the chains throughout

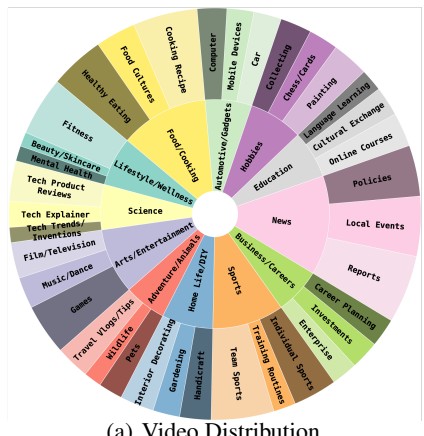
(a) Video Distribution

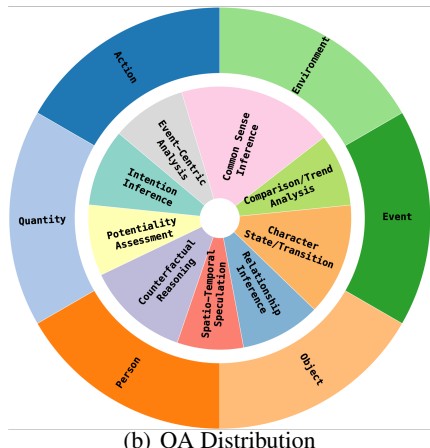
(b) QA Distribution

Figure 3: Distributions of videos and QA categories.

the duration of the video content. Identifying potential connections between two consecutive QA chains allows us to create a coherent narrative thread by adjusting the QA pairs in the following chain. This process helps to maintain a smooth multi-turn QA interaction across the entire video. The modified relation $\widetilde{r}_j$ is as follows:

$$\widetilde{r}_j = \langle Q_x^i, A_x^i, \widetilde{Q}_y^{i+1}, \widetilde{A}_y^{i+1}, \widetilde{Rc}\rangle, \tag{4}$$

where $\widetilde{Rc}$, $\widetilde{A}_y^{i+1}$ and $\widetilde{Q}_y^{i+1}$ respectively represent the modified relationship category, the question and answer in the subsequent chain after modification. During the modification phase, the following criteria are met: (1) Contextual coherence within each QA pair of a single chain (e.g., $\langle Q_{y-1}^{i+1}, A_{y-1}^{i+1}\rangle \rightarrow \langle \widetilde{Q}_y^{i+1}, \widetilde{A}_y^{i+1}\rangle \rightarrow \langle Q_{y+1}^{i+1}, A_{y+1}^{i+1}\rangle$) has to be preserved. (2) Logical links should be forged both within individual chains and between different chains, to facilitate cross-clip reasoning, as illustrated by $\langle Q_x^i, A_x^i\rangle \xrightarrow{\widetilde{Rc}} \langle \widetilde{Q}_y^{i+1}, \widetilde{A}_y^{i+1}\rangle$. (3) Through analytical reasoning and harnessing inter-clip information, repetitive, similar, and simple QAs should be changed into more in-depth and complex QAs. Given the difficulty of simultaneously meeting these stringent criteria, utilizing LLMs to establish these linkages proves to be impractical. Therefore, manual annotation is required, with the specific modification guideline outlined in Appendix C.

## 4 STATISTICAL ANALYSIS

We present a comprehensive statistical analysis of our dataset, which is the first dataset annotated with temporal multi-turn dialogues for streaming videos, to the best of our knowledge. Our dataset comprises 1,353 videos from 6 distinct sources and stands out with an average of 36.94 semi-automatically annotated QA pairs per video, which is the highest among existing datasets. Notably, it also consists of long videos with an average length exceeding 2 minutes. Moreover, each multi-turn dialogue in our dataset contains an average of 4.29 QA pairs and every video contains an average of 8.61 multi-turn dialogues. The details and comparisons are illustrated in Table 1.

**Video Categories.** Our dataset contains videos organized into 12 primary categories and 36 subcategories, as depicted in Figure 3(a), which illustrates the the diversity and inclusiveness of video types within our dataset.

**Question Categories**. To facilitate a more comprehensive evaluation of the capabilities of LVLMs, we classify the questions into 9 distinct categories as shown in Figure 3(b). Each category corresponds to the assessment for one specific skill of LVLMs. The criteria for these categories are as follows: (1) *Intention Inference (II)*: Discerning the underlying intention behind actions of characters. (2) *Potentiality Assessment (PA)*: Evaluating the feasibility of an action under certain conditions. (3) *Counterfactual Reasoning (CR)*: Analyzing outcomes by hypothesizing alternative scenarios. (4) *Spatio-Temporal Speculation (STS)*: Understanding the spatial and temporal relationships within the video. (5) *Relationship Inference (RI)*: Identifying and interpreting the relationships between entities. (6) *Character State and Transition (CST)*: Tracking the emotional states and transitions

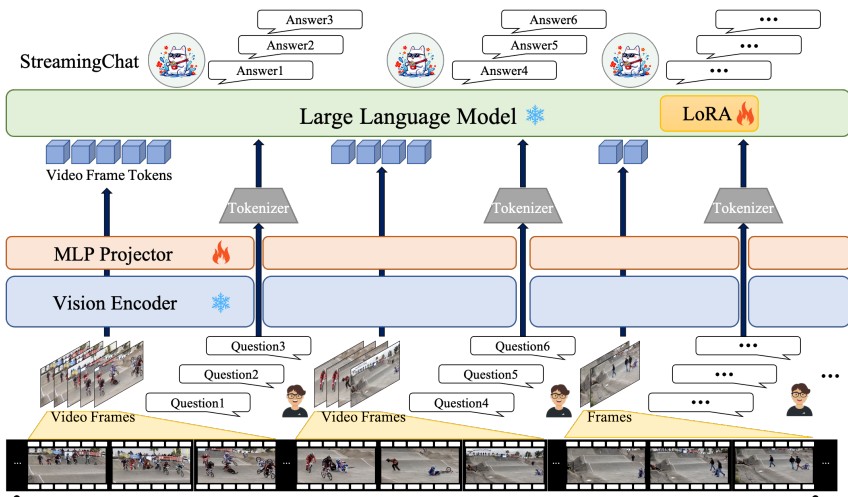

Figure 4: Architecture of the proposed StreamingChat model.

of characters under analysis within specific contexts. (7) *Comparison and Trend Analysis (CTA)*: Comparing different entities and analyzing emerging trends. (8) *Common Sense Inference (CSI)*: Applying general world knowledge to provide a logical framework. (9) *Event-Centric Analysis (ECA)*: Focusing on in-depth examination of significant events. Details of the above categories are provided in Appendix B.

## 5 STREAMINGCHAT

**Model Architecture.** Built upon InternVL2 Chen et al. (2024d), we develop a streaming LVLM baseline named StreamingChat. It comprises a vision encoder (InternViT Chen et al. (2023)), an MLP projector, and an LLM (InternLM2 Cai et al. (2024)), as illustrated in Figure 4. For the vision encoder, we employ the InternViT model (pre-trained on a combination of image captioning and OCR-specific datasets) to extract video frame embeddings at 1 FPS. To enhance the efficiency of learning streaming video understanding capabilities, we fine-tune the model using a static resolution strategy, which allows the model to handle several minutes of video and context within a 32k context window. The extracted frame embeddings are then fed into an MLP projector to generate frame tokens, following the approach used in LLaVA-1.5 Liu et al. (2024a). These frame tokens are interleaved with language tokens and input into the LLM, InternLM2. Additionally, we incorporate LoRA Hu et al. (2021) in every linear layer of the LLM to facilitate efficient tuning.

**Supervised Fine-Tuning Data.** We utilize the training data from our dataset for supervised fine-tuning to enhance the performance of StreamingChat. Each data entry represents a temporal dialogue path, which is converted into conversation data. The data format follows a multi-turn, multi-image structure. Specifically, the video multi-turn dialogue format involves sequentially inputting segments of the video and engaging in several rounds of dialogue for each segment until the entire video has been processed: *<video>Segment 1 </video><question_1><answer_1>... <question_N><answer_N><video>Segment 2 </video>...* Through this approach, we sequentially input video segments and engage in multiple rounds of dialogue for each segment until the entire video has been processed. Due to the context window limitation, we split temporal dialogue paths exceeding 100 frames into multiple segments for training.

## 6 EXPERIMENTS

### 6.1 EXPERIMENTAL SETUP

To effectively evaluate the performance of current LVLMs in streaming video understanding, we meticulously select a diverse range of state-of-the-art models, encompassing both open-source and closed-source LVLMs. We design two distinct experimental setups within the SVBench evaluation set to rigorously assess the capabilities of these LVLMs.

Table 2: Evaluation results of various models on SVBench in dialogue and streaming evaluation.

| Model | Dialogue Evaluation | | | | | | Streaming Evaluation | | | | | |
|---|---|---|---|---|---|---|---|---|---|---|---|---|
| | SA | CC | LC | TU | IC | OS | SA | CC | LC | TU | IC | OS |
| **Open-source LVLMs** | | | | | | | | | | | | |
| MovieChat | 20.46 | 20.05 | 27.76 | 21.81 | 22.21 | 21.89 | 17.99 | 16.42 | 20.37 | 15.77 | 19.08 | 17.43 |
| Video-ChatGPT | 31.86 | 32.58 | 40.28 | 35.32 | 36.26 | 33.80 | 27.98 | 29.54 | 33.81 | 27.95 | 31.00 | 28.88 |
| Video-LLaVA | 35.62 | 36.52 | 42.93 | 38.63 | 38.84 | 37.34 | 32.22 | 32.83 | 36.35 | 32.46 | 34.54 | 32.79 |
| ShareGPT4Video | 39.01 | 40.42 | 47.89 | 41.42 | 43.18 | 40.70 | 34.65 | 36.70 | 41.07 | 35.76 | 37.22 | 35.79 |
| VideoLLaMA2 | 39.13 | 40.33 | 47.60 | 42.36 | 41.80 | 40.60 | 35.68 | 36.40 | 42.23 | 34.65 | 36.70 | 35.84 |
| TimeChat | 36.19 | 37.06 | 44.72 | 40.42 | 37.12 | 37.22 | 35.72 | 37.88 | 42.65 | 36.23 | 36.34 | 36.32 |
| InternVL2 | 45.91 | 46.30 | 52.67 | 49.81 | 46.25 | 46.13 | 43.55 | 44.10 | 48.91 | 40.95 | 44.17 | 42.71 |
| VILA | 46.83 | 48.41 | 54.92 | 48.30 | 50.12 | 48.51 | 46.19 | 47.95 | 51.60 | 44.84 | 48.56 | 46.26 |
| InternLM-XComposer2.5 | 51.57 | 53.93 | 59.69 | 51.57 | 56.28 | 52.31 | 52.22 | 53.39 | 58.14 | 48.05 | 54.79 | 51.46 |
| MiniCPM-V 2.6 | 53.50 | 55.42 | 60.88 | 55.03 | 55.78 | 54.30 | 53.33 | 54.30 | 58.97 | 49.64 | 54.71 | 52.19 |
| StreamingChat | **59.48** | **61.31** | **66.05** | **58.61** | **61.09** | **59.41** | **55.10** | **56.66** | **60.72** | **51.78** | **55.87** | **53.90** |
| **Closed-source LVLMs** | | | | | | | | | | | | |
| Gemini 1.5 Pro | 54.89 | 56.05 | 61.45 | 53.08 | 56.06 | 54.29 | 49.06 | 50.05 | 54.62 | 45.73 | 49.84 | 48.02 |
| GPT-4V | 65.56 | 68.02 | 71.78 | 63.80 | 68.01 | 65.19 | 58.82 | 59.55 | 64.29 | 54.08 | 60.61 | 57.35 |
| GPT-4o | **65.73** | **68.10** | **71.95** | **66.54** | **68.40** | **66.29** | **59.52** | **60.42** | **65.45** | **55.10** | **61.36** | **58.17** |

**Dialogue Evaluation.** In this setup, we evaluate the capabilities of LVLMs in understanding long-context scenarios within temporal multi-turn dialogues. Each LVLM is provided with a contextual history that includes all preceding QA pairs up to the current timestamp in the QA chain. Once a dialogue sequence concludes chronologically, the model transitions to the next video clip and addresses its associated QA chain. This process continues until the entire video is played and every question has been answered. This evaluation allows us to assess the ability of LVLMs to maintain continuity over multiple turns and to respond while considering the accumulated context. The aim is to simulate real-world scenarios where users pose a series of related questions while watching a video, requiring the model to track and integrate information over time.

**Streaming Evaluation.** Building upon the dialogue evaluation, this setup focuses on assessing the ability of LVLMs to perform temporal reasoning by introducing probabilistic transitions between related QA chains. Similar to the above setup, models are presented with the context up to the current timestamp. However, when encountering questions that have temporal linkages to subsequent QA chains, there is an 80% probability that the model will jump to the corresponding related question in the following chain. This evaluation aims to challenge the understanding of temporal dependencies and its capability to reason about the sequence of events across different but related video segments.

## 6.2 EVALUATION METRICS

**Basic Metrics.** Evaluating performance in streaming video understanding requires sophisticated metrics to capture various dimensions. Below, we outline the commonly-used evaluation metrics.

- **METEOR** Banerjee & Lavie (2005) evaluates the precision, recall, and alignment of words and phrases between the references and the ground truth by considering synonymy and stemming.
- **GPT4-Score** assessed by GPT-4, evaluates the accuracy of generated answers solely based on the semantic similarity between a single answer and the ground truth.

**Dialogue Evaluation Framework.** To evaluate the capabilities in both dialogue and streaming evaluations, it is crucial to adopt a multidimensional framework that assesses the quality of multi-turn dialogues. We propose an LLM-based evaluation framework encompassing several key aspects that contribute to the holistic assessment of LVLMs. See Appendix H for detailed prompts.

- **Semantic Accuracy (SA)** evaluates the accuracy of the generated answers based on a holistic understanding. It considers not only the direct overlap with ground-truth answers but also the context, coherence, and overall relevance of the response to the question posed.
- **Contextual Coherence (CC)** examines the ability to maintain relevance and context across sequential questions and answers, ensuring continuity and alignment with the evolving discourse.
- **Logical Consistency (LC)** evaluates the logical progression and consistency of answers, ensuring that answers do not contradict each other or previous information.

Table 3: Evaluation results on SVBench across 9 long-context streaming video understanding skills.

| Model | II | PA | RI | CR | CST | CSI | CTA | ECA | STS |
|---|---|---|---|---|---|---|---|---|---|
| **Open-source LVLMs** | | | | | | | | | |
| MovieChat Song et al. (2024) | 21.91 | 21.32 | 18.82 | 17.78 | 24.58 | 19.23 | 16.77 | 16.59 | 15.41 |
| Video-ChatGPT Maaz et al. (2023) | 35.85 | 40.12 | 31.18 | 29.37 | 39.38 | 29.95 | 25.77 | 26.97 | 22.79 |
| TimeChat Ren et al. (2024) | 33.26 | 43.62 | 29.86 | 30.58 | 39.97 | 28.49 | 27.08 | 28.10 | 24.22 |
| VideoLLaMA2 Cheng et al. (2024) | 36.76 | 47.38 | 30.56 | 33.08 | 39.85 | 37.09 | 29.67 | 31.75 | 26.03 |
| ShareGPT4Video Chen et al. (2024b) | 39.92 | 47.54 | 33.61 | 33.56 | 43.69 | 35.97 | 28.98 | 31.25 | 26.43 |
| Video-LLaVA Lin et al. (2023) | 38.99 | 46.42 | 35.86 | 34.27 | 46.09 | 35.65 | 31.21 | 31.20 | 26.67 |
| InternVL2 Chen et al. (2024d) | 37.10 | 51.19 | 38.25 | 36.79 | 36.02 | 41.66 | 32.22 | 33.24 | 29.07 |
| VILA Lin et al. (2024) | 42.13 | 55.48 | 40.60 | 39.48 | 45.93 | 43.29 | 35.87 | 34.22 | 31.21 |
| InternLM-XComposer2.5 Zhang et al. (2024b) | 47.64 | 57.71 | 40.49 | 41.72 | 50.00 | 47.20 | 36.99 | 39.53 | 33.90 |
| MiniCPM-V 2.6 Yao et al. (2024) | 43.25 | 50.45 | 39.93 | 39.96 | 51.61 | 47.40 | 37.31 | 38.67 | 33.43 |
| StreamingChat (Ours) | **53.94** | **71.96** | **50.22** | **50.49** | **59.26** | **53.46** | **44.60** | **47.68** | **37.99** |
| **Closed-source LVLMs** | | | | | | | | | |
| Gemini 1.5 Pro Reid et al. (2024) | 41.29 | 43.32 | 42.24 | 39.10 | 47.20 | 50.80 | 39.19 | 34.78 | 35.98 |
| GPT-4V Yang et al. (2023) | 56.84 | **61.57** | 49.10 | **51.31** | 58.57 | 55.76 | 47.47 | 47.44 | 42.97 |
| GPT-4o Achiam et al. (2023) | **57.95** | 59.47 | **52.29** | 49.97 | 56.63 | **58.65** | **49.02** | 47.47 | **44.58** |

- **Temporal Understanding (TU)** assesses the model's proficiency in comprehending and reasoning about temporal events and sequences depicted in the video content.

- **Informational Completeness (IC)** measures the comprehensiveness to gauge whether the model captures and conveys all relevant elements from the video to provide a thorough answer.

- **Overall Score (OS)** is derived by aggregating the scores from each aforementioned criterion.

## 6.3 OVERALL PERFORMANCE

The dialogue and streaming evaluation results on SVBench, outlined in Table 2, provide a comprehensive comparison among various LVLMs. Notably, closed-source models such as GPT-4o and GPT-4V attain significantly higher scores across all metrics, with GPT-4o achieving an Overall Score (OS) of 66.29 in dialogue evaluation and 58.17 in streaming evaluation. Among the open-source models, StreamingChat and MiniCPM-V 2.6 stand out as top performers, achieving OS scores of 59.41 and 54.30 respectively in dialogue evaluation, and 53.90 and 52.19 in streaming evaluation. Notably, StreamingChat demonstrates a significant improvement over the original InternVL2, with a 28.79% increase in dialogue evaluation and a 26.20% increase in streaming evaluation. This underscores the effectiveness of SVBench training data for streaming video understanding tasks.

Additionally, we conduct performance comparisons before and after fine-tuning on 6 image and video understanding benchmarks, as illustrated in Figure 5. The results indicate that the fine-tuned StreamingChat shows substantial improvements over the original InternVL2 on SVBench. While there are slight decreases in performance on the image benchmark MMBench Liu et al. (2023) and the video benchmark MMBench-Video Fang et al. (2024), there are modest gains on the video benchmarks VideoMME Fu et al. (2024) and MVBench Li et al. (2024). These findings suggest that StreamingChat enhances streaming video understanding capabilities without compromising fundamental image and video comprehension skills.

It is evident that scores in streaming evaluation are consistently lower compared to those in dialogue evaluation. This discrepancy can be attributed to the inherent complexity of streaming evaluation, which demands seamless comprehension and processing of dynamically evolving video content. Unlike dialogue evaluation, which deals with relatively static and contextually stable inputs, streaming evaluation necessitates understanding and integration of extended temporal contexts and dynamic scenes, posing significant challenges for current models.

## 6.4 PERFORMANCE ANALYSIS OF VIDEO UNDERSTANDING SKILLS

The analysis of model performance across 9 long-context streaming video understanding skills (see Section 4) in SVBench reveals significant disparities, underscoring the inherent challenges posed by different types of skills. According to Table 3, skills such as Intention Inference (II) and Potentiality Assessment (PA) generally exhibit higher performance across most models, particularly in closed-source LVLMs such as GPT-4V and GPT-4o. This suggests that both open-source and closed-source models are relatively adept at deciphering character intentions and predicting potential future

Table 4: Ablation study on single-instance (Sin.) and multi-turn (Mul.) QA evaluation.

| Model | METEOR | | GPT4-Score | |
|---|---|---|---|---|
| | Sin. | Mul. | Sin. | Mul. |
| **Open-source LVLMs** | | | | |
| MovieChat | 19.20 | 23.77 | 17.18 | 21.81 |
| Video-ChatGPT | 25.93 | 30.44 | 26.83 | 34.62 |
| Video-LLaVA | 26.29 | 31.51 | 30.65 | 40.03 |
| ShareGPT4Video | 26.31 | 31.38 | 30.41 | 39.68 |
| VideoLLaMA2 | 24.52 | 30.87 | 29.67 | 38.88 |
| TimeChat | 25.50 | 27.47 | 26.64 | 34.42 |
| InternVL2 | 26.85 | 32.74 | 32.26 | 42.25 |
| VILA | 28.44 | 33.57 | 34.63 | 45.41 |
| InternLM-XComposer2.5 | 23.32 | 31.02 | 37.11 | 49.69 |
| MiniCPM-V 2.6 | 27.36 | 33.90 | 36.73 | 48.57 |
| StreamingChat | **32.94** | **35.58** | **43.85** | **57.94** |
| **Closed-source LVLMs** | | | | |
| Gemini 1.5 Pro | 26.63 | 32.91 | 36.85 | 48.83 |
| GPT-4V | 28.66 | 36.47 | 44.94 | 60.11 |
| GPT-4o | **28.81** | **36.84** | **45.37** | **60.70** |

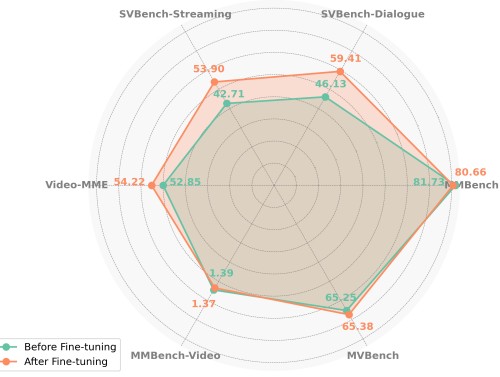

Figure 5: Performance comparisons on 6 image and video understanding benchmarks: Before fine-tuning (InternVL2) and after fine-tuning (StreamingChat).

actions. Conversely, more complex skills like Counterfactual Reasoning (CR) and Spatio-Temporal Speculation (STS) manifest comparatively lower accuracy. This trend is echoed in closed-source models, where even the high-performing GPT-4V and GPT-4o models show relative declines in performance, though still maintaining superior scores compared to other models. This drop can be attributed to the cognitive demands of these tasks, which involve abstract reasoning, intricate scenario construction, and temporal-spatial awareness—areas that current models struggle to emulate effectively. RI and ECA also pose significant challenges (see Appendix H). Both involve multi-entity and context-sensitive interactions, leading to performance variability. Notably, StreamingChat outperforms other open-source LVLMs across all 9 skills, and even surpasses closed-source models like GPT in PA, CST, and ECA. Nearly all models scored below 60 on 9 long-context streaming video understanding skills, suggesting that SVBench poses a challenging task.

## 6.5 ABLATION STUDY

Ablation studies are conducted to assess the effectiveness of dialogue evaluation (multi-turn QA denoted as "Mul.") versus traditional evaluation (single-instance QA denoted as "Sin.") in our dataset. As shown in Table 4, models generally exhibit improved metrics on both METEOR and GPT4-Score when additional contextual information from previous QAs is incorporated. This trend is consistent across both open-source and closed-source LVLMs. These results indicate that dialogue evaluation can significantly enhance the performance of models when applied to streaming video understanding. Unlike the traditional single-instance QA evaluation, the dialogue evaluation leverages the rich contextual information accumulated from previous interactions, thereby providing a more comprehensive evaluation framework. Despite the overall observed improvements in performance enabled by dialogue evaluation, there are notable instances where models exhibit relatively low scores even in this enhanced setting. These instances provide valuable insights into the limitations and areas for improvement in existing frameworks for streaming video understanding.

## 7 CONCLUSION

This paper presents **SVBench**, a novel benchmark for the assessment of long-context streaming video understanding. SVBench comprises a diverse collection of 1,353 streaming videos from 6 streaming platforms and 49,979 meticulously annotated question-answer pairs for temporal multi-turn dialogues. Our experiments reveal that while state-of-the-art LVLMs have made strides in single-instance video QA, their performance on streaming videos falls short of human-level accuracy. Motivated by this, we develop a StreamingChat model, which significantly outperforms open-source LVLMs on our SVBench and achieves comparable performance on diverse vision-language benchmarks. By providing a challenging benchmark, we hope to stimulate the development of advanced models capable of tackling the complexities of streaming video understanding.

## 8 ACKNOWLEDGMENT

We would like to present our appreciation to the anonymous reviewers and ACs for their constructive suggestions. This work is supported by the National Key Research and Development Program of China (No.2023YFC3310700), the Beijing Natural Science Foundation (JQ23018), and the National Natural Science Foundation of China (No.62036012, 62276257).

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

# Appendix

## A  DETAILS OF DATA COLLECTION

### A.1  DATA FILTERING

Initially, we source raw video data from a variety of publicly available datasets, including YT-Temporal-1B, YouCook2, ActivityNet, MovieChat, Panda-70M, and Ego4D. These datasets offer a wide spectrum of video content, ensuring a rich diversity that is essential for streaming video understanding. In our initial filtering stage, we enforce a minimum duration threshold, excluding videos shorter than 1 minute to ensure sufficient temporal information depth. Subsequently, we employ aesthetic assessments to retain videos with an aesthetic score of 4 or above, thereby ensuring video clarity and visual quality. Furthermore, we filter videos with optical flow scores within the range of 0.5 to 100, capturing both visual complexity and motion coherence.

## A.2 AESTHETIC ASSESSMENT

To further ensure video clarity and visual quality, we apply Open-Sora to assign a score for the aesthetic appeal of videos. Videos with an aesthetic score of 4 or above are retained. This criterion ensures that videos within our dataset are not only content-rich videos but also with high aesthetic appeal.

## A.3 OPTICAL PERFORMANCE ASSESSMENT

Furthermore, we apply Open-Sora for assigning optical flow scores to each video within our dataset as well, which reflects both the visual complexity and the motion coherence of a video. We filter videos with optical flow scores within the range of 0.5 to 100 to ensure the quality of videos within our dataset.

## A.4 SCENE DETECTION AND VIDEO SPLITTING

After the filtering and the assessment above, we employ advanced scene detection algorithms to identify and enumerate scenes within each video. These results are crucial in determining whether the video content exhibits adequate variation and complexity. Further filtering is conducted to retain only those videos containing 5 to 15 scenes, thereby excluding content that is either excessively monotonous or overly intricate. Moreover, we calculate the average duration of scenes for each video. Only videos with an average scene duration between 5 and 30 seconds are chosen, ensuring fluidity and rhythm that are essential for effective streaming data analysis. Ultimately, we split each video into clips based on timestamps, merging any clips shorter than 2 seconds with their adjacent ones. Notably, to prevent disjointedness and ensure continuity between scenes, we add an extra 0.5 seconds to both the beginning and end of each clip, resulting in a one-second overlap between consecutive clips.

## A.5 DISTRIBUTION OF VIDEO LENGTHS

Figure 6 shows the distribution of video lengths. The results indicate that 95.05% of the videos in the dataset are longer than 1 minute, primarily ranging from 60 to 240 seconds.

# B QUESTION CATEGORIES

All the questions within our dataset fall into nine categories as follows, with each category corresponding to the assessment for one specific skill of LVLMs.

## B.1 INTENTION INFERENCE

A category aimed at delving into the underlying intent behind an action. It assesses whether LVLMs truly comprehend the latent motivation behind an action, rather than simply observing the action itself.

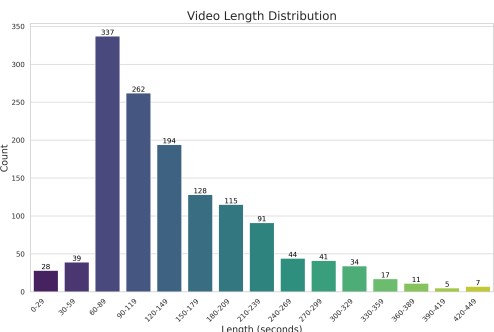

Figure 6: Distribution of video lengths.

## B.2 POTENTIALITY ASSESSMENT

A category that involves inferring the feasibility of an action under certain conditions or its potential outcomes if the context is changed. It evaluates whether LVLMs understand the nature of an action and the conditions required for realizing an action.

### B.3 COUNTERFACTUAL REASONING

A category that entails hypothetical reasoning within a scenario opposite to the existing facts. It assesses the capacity of LVLMs for causal understanding and recognizing the significance of variables.

### B.4 SPATIO-TEMPORAL SPECULATION

A category concerned with placing events or actions in particular temporal and spatial contexts to infer their implications, differences, or changes. It assesses the capacity of LVLMs to comprehend the influence of temporal and spatial factors on entities.

### B.5 RELATIONSHIP INFERENCE

A category focusing on exploring the links or associations between entities (such as persons or objects). It evaluates the capacity of LVLMs to understand the nature of relationships on the basis of observed interactions.

### B.6 CHARACTER STATE AND TRANSITION

A category that involves dissecting emotional states and transitions of characters under analysis within specific contexts. It gauges the capacity of LVLMs to accurately perceive emotional states and the potential impact of contextual changes.

### B.7 COMPARISON AND TREND ANALYSIS

A category dedicated to the comparison and trends across different entities (objects, actions, or events). It assesses the capacity of LVLMs to accurately discern similarities and differences of different entities and analyse the changes in trends.

### B.8 COMMON SENSE INFERENCE

A category that employs common sense or established facts to provide a logical framework for contextual inference. It evaluates the capacity of LVLMs to understand the essences of entities and utilize common sense for accurate inference.

### B.9 EVENT-CENTRIC ANALYSIS

A category devoted to in-depth examination of significant events, including investigation of triggering factors and their effects on the overall narrative trajectory. It evaluates the capacity of LVLMs to comprehend the nature and future trajectories of the events.

## C MODIFICATION GUIDELINE

In the modification phase, adherence to specific criteria is essential. To this end, we propose a comprehensive modification guideline.

Due to our categorization of QA pairs into six types of relationships, the modification guideline we propose is also divided into six major modules, including Action, Person, Object, Event, Environment, and Quantity. Each module contains more detailed methods of modification to assist annotators in revising questions to better evaluate diverse skills of LVLMs.

### C.1 ACTION

If adjacent chains contain QA pairs related to action, QA pairs within the subsequent chain can be modified according to the methods mentioned in the Action module. The specific method of modification is determined by whether the actions involved in the QA pairs within the adjacent chains are the same.

Table 5: Results of Different Primary Evaluation Metrics on Various LVLMs

| Model | BLEU-4 | METEOR | ROUGE-L | CIDEr | GPT4-Score |
|---|---|---|---|---|---|
| **Open-source LVLMs** | | | | | |
| MovieChat | 1.87 | 23.77 | 17.74 | 14.68 | 21.81 |
| Video-ChatGPT | 4.69 | 30.44 | 26.17 | 51.60 | 34.62 |
| Video-LLaVA | 5.38 | 31.51 | 26.61 | 62.84 | 40.03 |
| ShareGPT4Video | 4.51 | 31.38 | 26.64 | 52.60 | 39.68 |
| VideoLLaMA2 | 4.43 | 30.87 | 24.40 | 49.07 | 38.88 |
| TimeChat | 4.56 | 27.47 | 25.63 | 64.52 | 34.42 |
| InternVL2 | 5.96 | 32.74 | 27.32 | 67.57 | 42.25 |
| VILA | 6.37 | 33.57 | 29.30 | 75.31 | 45.41 |
| InternLM-XComposer-2.5 | 4.36 | 31.02 | 22.90 | 39.18 | 49.69 |
| MiniCPM-V 2.6 | 5.94 | 33.90 | 27.95 | 73.97 | 48.57 |
| StreamingChat | **8.30** | **37.58** | **34.92** | **91.11** | **57.88** |
| **Closed-source LVLMs** | | | | | |
| Gemini 1.5 Pro | 5.78 | 32.91 | 27.03 | 78.51 | 48.83 |
| GPT-4V | 6.67 | 36.47 | 29.58 | 82.45 | 60.11 |
| GPT-4o | **6.93** | **36.84** | **29.77** | **83.69** | **60.70** |

If the actions are the same (Q1 is similar to Q2 and A1 is similar to A2), the following modification methods can be adopted:

- **Modify the question based on the purpose of the action.** Change Q2 to "What is the purpose or intent of this action?"

- **Modify the question based on the likelihood of the action occurring.** Change Q2 to "Is this action possible under the given conditions? If so, what conditions need to be met?"

- **Modify the question based on the purpose of the action and spatio-temporal inference.** Change Q2 to "What is the purpose or intent of this action at a specific time and place?"

- **Modify the question based on counterfactuals.** Change Q2 to "What different outcomes or impacts might result from taking a different action?"

- **Modify the question based on counterfactuals and spatio-temporal inference.** Change Q2 to "What different outcomes or impacts might result if this action were performed at another time and place?"

- **Modify the question based on the sequence of actions.** Change Q2 to "What is the action before/after this one?" or further modify Q2 to "How does the action before this one affect the execution of the current action?" to deepen the inquiry.

- **Modify the question based on the comparison of multiple actions.** Change Q2 to "Which action at a specific time and place is more complex/simple compared to a specific action at a previous time and place?" or further modify Q2 to "How long is the interval between these two actions in time?" to deepen the inquiry.

If the actions are different (Q1 is not similar to Q2 and A1 is not similar to A2), the following modification methods can be adopted:

- **Modify the question based on sequence.** Change Q2 to "What action is taken next?"

## C.2 QUANTITY

If adjacent chains contain QA pairs related to quantity, QA pairs within the subsequent chain can be modified according to the methods mentioned in the quantity module, as follows:

- **Modify the question based on the comparison of two quantities.** Change Q2 to "Comparing the quantities of two categories/types of things, which is more/less?" For example,

if Q1 is "How many people are involved on the field?" and Q2 is "How many red balls are on the field?" change Q2 to "Which is greater in number on the field, the balls or the people?"

- **Modify the question based on the comparison of two quantities and the trend of change.** Change Q2 to "What is the trend of change in the quantity of these things under a certain context/time?" or further modify Q2 to "Does this change have periodicity or regularity, and if so, what is the nature of this regularity?" to deepen the inquiry.

## C.3 PERSON

If adjacent chains contain QA pairs related to people, QA pairs within the subsequent chain can be modified according to the methods mentioned in the person module, and the specific method of modification is determined by whether the people involved in the QA pairs within the adjacent chains are the same.

If the people involved are different (P2 is different from P1), the following modification methods can be adopted:

- **Modify the question based on the social relationship between two people.** Change Q2 to "From the existing video, what is the relationship between P2 and P1?" or further modify Q2 to "Does their relationship change in different plots of the video?" to deepen the inquiry.
- **Modify the question based on the relative positional relationship between two people.** Change Q2 to "In the most recent scene where P1 and P2 appear together, what is the positional relationship between P2 and P1?" or further modify Q2 to "How does the distance and direction between them change in a certain context/time?" to deepen the inquiry.
- **Modify the question based on the interaction between two people.** Change Q2 to "What is the interaction between characters in a certain context/time?"

If the people involved are the same (P2 is the same as P1), the following modification methods can be adopted:

- **Modify the question based on the emotional state of the person.** Change Q2 to "Based on a specific context/time in the video, how does this person's emotional state change?" or further modify Q2 to "How does the change in emotional state affect behavior?" to deepen the inquiry.
- **Modify the question based on the person's identity background.** Change Q2 to "What is the background of this person?"

## C.4 OBJECT

If adjacent chains contain QA pairs related to the same object, QA pairs within the subsequent chain can be modified according to the methods mentioned in the object module, as follows:

- **Modify the question based on inference.** Change Q2 to "How does the existence or characteristics of this object affect other objects or events?" or further modify Q2 to "If the object no longer exists or its characteristics change, how would the event differ?" to deepen the inquiry.
- **Modify the question based on counterfactuals.** Change Q2 to "If certain attributes or characteristics of this object were different, what different outcomes or impacts might there be?"
- **Modify the question based on outcomes.** Change Q2 to "What is the contribution of this object to a certain event or phenomenon?"
- **Modify the question based on counterfactuals and spatio-temporal inference.** Change Q2 to "If this object were not present at a certain time and place for a certain event or phenomenon, what different impact or outcome might there be?"
- **Modify the question based on impact and spatio-temporal inference.** Change Q2 to "What is the contribution of this object to a certain event or phenomenon at a specific time and place?"

- **Modify the question based on counterfactuals and general knowledge.** Change Q2 to "If certain attributes or characteristics of this object were different, how might it affect our common sense or known facts?"

- **Modify the question based on impact and general knowledge.** Change Q2 to "How does the existence or characteristics of this object affect our common sense or known facts?"

- **Modify the question based on state evolution.** Change Q2 to "How does the state of this object change in a certain context/time in the video?"

- **Modify the question based on the trajectory of action.** Change Q2 to "What is the trajectory of this object's action in a certain context/time in the video?"

## C.5 EVENT

If adjacent chains contain QA pairs related to the same event, QA pairs within the subsequent chain can be modified according to the methods mentioned in the event module, as follows:

- **Modify the question based on sequence.** Change Q2 to "What happens next?"

- **Modify the question based on the relationship of events.** Change Q2 to "What event caused this event?" or further modify Q2 to "What kind of chain reaction did the occurrence of this event cause?" to deepen the inquiry.

- **Modify the question based on event classification.** Change Q2 to "To which category of events does this event belong (e.g., natural disasters, social events, etc.)?"

- **Modify the question based on the impact of events.** Change Q2 to "What impact will this event have?"

- **Modify the question based on event prediction.** Change Q2 to "Based on the current situation, what events might occur in the future?" or further modify Q2 to "Based on the current situation, how likely is it that certain events will occur in the future, and what is the basis for this?" to deepen the inquiry.

## C.6 ENVIRONMENT

If adjacent chains contain QA pairs related to the same environment, QA pairs within the subsequent chain can be modified according to the methods mentioned in the environment module, as follows:

- **Modify the question based on environmental changes and trends.** Change Q2 to "Compared to before, what changes are there in the current environment?" or further modify Q2 to "How will this environment change afterward?" to deepen the inquiry.

- **Modify the question based on counterfactuals.** Change Q2 to "Without this environment, what impact would there be?"

## D PRIMARY EVALUATION METRICS

As shown in the Table 5, we employ four primary evaluation metrics to assess the answers generated by various LVLMs mentioned and compare the evaluation results with those evaluated by GPT-4. It can be observed that the results from these four primary evaluation metrics are broadly in agreement with the evaluation made by GPT-4, which suggests that our SVBench is capable of accurately and effectively distinguishing distinct levels of capabilities possessed by various LVLMs. Just like the use of sentence similarity Di & Xie (2024) in QAEgo4D as an important evaluation metric for computing distances in semantic space, we will subsequently use semantic similarity as an evaluation criterion to assess the consistency between the model's output answers and the ground truth.

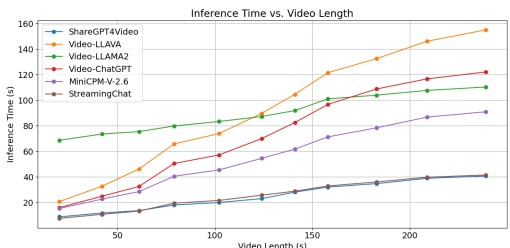 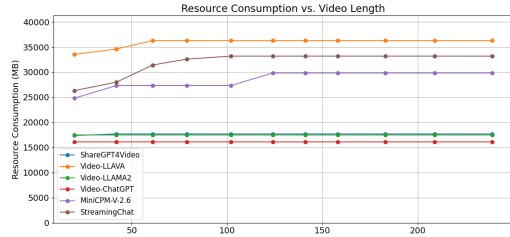

Figure 8: Correlation between inference speed with video length.

Figure 9: Correlation between resource consumption with video length.

# E  CASE STUDY

## E.1  BENCHMARK CASE

We employ qualitative comparisons in the necessity of contextual content and the responses within a QA chain between our SVBench and answers generated by both GPT-4V and GPT-4o, with details in both the Figure 11 and the Figure 12. Furthermore, we conduct additional comparisons in streaming question-answering and nine skills assessment between our SVBench and answers generated by GPT-4o, as shown in Figure 13 and Figure 14. These comparisons above present that our SVBench is a challenge to the existing LVLMs.

## E.2  MODEL CASE

As demonstrated by Figure 15, the current best-performing open-source model, MiniCPM-V 2.6, is unable to analyze the questioner's intent based on contextual information and provide the most appropriate answer. MiniCPM-V 2.6 relies on the most accurate information from the current question and video segments to answer as accurately as possible (and fails to respond when information is insufficient), but it overlooks the coherence and contextual information between questions.

# F  ADDITIONAL EXPERIMENTS

## F.1  COMPARISON OF LLM-BASED EVALUATIONS WITH HUMAN EVALUATION

To ensure consistency with LLM assessments, we incorporate human involvement in the generation process and human evaluation. We add human evaluations of various LVLMs results (see Figure 7). To validate the consistency between human scores and those from open-source and closed-source models, we employ human evaluation (10 people annotating 200 videos over a week), open-source model evaluation (InternVL2-Llama3-76B), closed-source evaluation model (GPT-4), and traditional evaluation metrics (METEOR). We then plot a score comparison on Multi-turn QA Evalua-

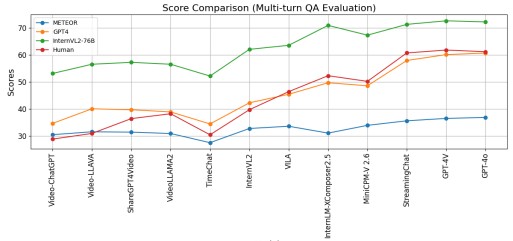

Figure 7: Score comparison on multi-turn QA evaluation (Mul.).

tion. The results indicate: (1) The score variations between Human and GPT-4 are generally consistent across different models, though human scores are more discriminative, suggesting that GPT-4 scores are reasonably reliable; (2) InternVL2-Llama3-76B shows excessive leniency, with more than half of the models scoring above 60 using the same prompt as GPT-4, while METEOR scores are too low, lacking discrimination.

Regarding the circular dependency issue, we mitigate the reliance on LLMs by incorporating two stages of manual annotations during the construction of annotations. Additionally, the inclusion of open-source and human evaluations enriches the evaluation results, further reducing the dependency on GPT-4 for evaluation.

Table 6: Evaluation results of various models on SVBench in dialogue and streaming evaluation.

| Model | Dialogue Evaluation | | | | | | Streaming Evaluation | | | | | |
|---|---|---|---|---|---|---|---|---|---|---|---|---|
| | SA | CC | LC | TU | IC | OS | SA | CC | LC | TU | IC | OS |
| Human | **88.43** | **87.47** | **89.58** | **88.29** | **76.15** | **83.93** | **83.57** | **80.71** | **87.86** | **86.43** | **75.71** | **80.24** |
| **Open-source LVLMs** | | | | | | | | | | | | |
| MovieChat | 20.46 | 20.05 | 27.76 | 21.81 | 22.21 | 21.89 | 17.99 | 16.42 | 20.37 | 15.77 | 19.08 | 17.43 |
| Video-ChatGPT | 31.86 | 32.58 | 40.28 | 35.32 | 36.26 | 33.80 | 27.98 | 29.54 | 33.81 | 27.95 | 31.00 | 28.88 |
| Video-LLaVA | 35.62 | 36.52 | 42.93 | 38.63 | 38.84 | 37.34 | 32.22 | 32.83 | 36.35 | 32.46 | 34.54 | 32.79 |
| ShareGPT4Video | 39.01 | 40.42 | 47.89 | 41.42 | 43.18 | 40.70 | 34.65 | 36.70 | 41.07 | 35.76 | 37.22 | 35.79 |
| VideoLLaMA2 | 39.13 | 40.33 | 47.60 | 42.36 | 41.80 | 40.60 | 35.68 | 36.40 | 42.23 | 34.65 | 36.70 | 35.84 |
| Flash-VStream | 38.23 | 41.56 | 51.51 | 45.99 | 43.32 | 42.69 | 36.78 | 37.50 | 40.48 | 32.46 | 35.23 | 35.92 |
| TimeChat | 36.19 | 37.06 | 44.72 | 40.42 | 37.12 | 37.22 | 35.72 | 37.88 | 42.65 | 36.23 | 36.34 | 36.32 |
| InternVL2 | 45.91 | 46.30 | 52.67 | 49.81 | 46.25 | 46.13 | 43.55 | 44.10 | 48.91 | 40.95 | 44.17 | 42.71 |
| LLaVA-NeXT-Video | 43.56 | 48.33 | 51.79 | 48.54 | 47.40 | 45.48 | 40.79 | 48.84 | 54.61 | 43.03 | 48.32 | 45.36 |
| VILA | 46.83 | 48.41 | 54.92 | 48.30 | 50.12 | 48.51 | 46.19 | 47.95 | 51.60 | 44.84 | 48.56 | 46.26 |
| Qwen2-VL | 48.52 | 50.65 | 54.75 | 43.04 | 48.59 | 48.65 | 48.94 | 52.46 | 53.24 | 48.32 | 50.70 | 49.48 |
| InternLM-XComposer2.5 | 51.57 | 53.93 | 59.69 | 51.57 | 56.28 | 52.31 | 52.22 | 53.39 | 58.14 | 48.05 | 54.79 | 51.46 |
| MiniCPM-V 2.6 | 53.50 | 55.42 | 60.88 | 55.03 | 55.78 | 54.30 | 53.33 | 54.30 | 58.97 | 49.64 | 54.71 | 52.19 |
| StreamingChat | **59.48** | **61.31** | **66.05** | **58.61** | **61.09** | **59.41** | **55.10** | **56.66** | **60.72** | **51.78** | **55.87** | **53.90** |
| **Closed-source LVLMs** | | | | | | | | | | | | |
| Gemini 1.5 Pro | 54.89 | 56.05 | 61.45 | 53.08 | 56.06 | 54.29 | 49.06 | 50.05 | 54.62 | 45.73 | 49.84 | 48.02 |
| GPT-4V | 65.56 | 68.02 | 71.78 | 63.80 | 68.01 | 65.19 | 58.82 | 59.55 | 64.29 | 54.08 | 60.61 | 57.35 |
| GPT-4o | **65.73** | **68.10** | **71.95** | **66.54** | **68.40** | **66.29** | **59.52** | **60.42** | **65.45** | **55.10** | **61.36** | **58.17** |

## F.2 CORRELATION BETWEEN INFERENCE SPEED AND CONSUMPTION WITH VIDEO LENGTH

Figure 8 and Figure 9 show the inference speed and resource consumption of multiple LVLMs in relation to video length. The inference time is generally positively correlated with the video length. For resource consumption, some models show a plateau as the input length increases, due to the saturation of input frames and the memory reaching its preset limit.

## F.3 ADDITIONAL EVALUATION RESULTS

We include a comparison with human performance, as shown in 6, which can provide valuable insights into the gap between current models and human capabilities in long-context streaming video understanding. The results show that human performance significantly outperforms various open-source and closed-source models across all metrics in both evaluation settings (Dialogue Evaluation and Streaming Evaluation). Additionally, humans excel in Temporal Understanding (TU) but perform relatively weaker in Informational Completeness (IC). We also included LVLMs such as Flash-VStream-7B Zhang et al. (2024a), Qwen2-VL-7B Wang et al. (2024a), and LLaVA-NeXT-Video-7B-DPO Zhang et al. (2024c). We will also add the evaluation results for the models (Oryx Liu et al. (2024b), Long-LLaVA Wang et al. (2024c), LongVILA Xue et al. (2024)) to Table 6 as soon as possible.

## F.4 IMPACT OF MODEL SIZE ON PERFORMANCE

To ensure a fair comparison, we select base models with 7B or 8B parameters. Our benchmark is an ongoing project, and we plan to update it with different sizes of mainstream Video-LLMs, maintaining a real-time updated leaderboard. Additionally, we conduct further experiments comparing the performance of models with different LLM sizes, specifically InternVL2-2B, 4B, 8B, 26B, 40B, and 76B, on SVBench, as shown in Figure 10. The results indicate that model size significantly impacts performance, with larger models demonstrating better accuracy. Interestingly, the performance of InternVL2 with 76B parameters is actually lower compared to the 40B parameter version.

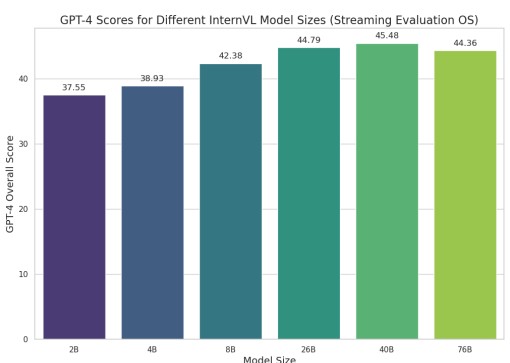

Figure 10: Impact of model sizes on performance of InternVL2.

Both models use the same vision part, InternViT-6B, but the LLM has been changed from Yi-based to Llama3-based. This suggests that Llama3 may have weaker visual language processing capabilities, as we also observed a similar decrease in performance on MVBench.

## G  REAL-TIME LEADERBOARD

Our benchmark is an ongoing project, and we plan to continuously update it with various mainstream Video-LLMs, maintaining a real-time updated leaderboard. Given the resource-intensive nature of evaluating a large number of newly emerging models using GPT-4o, we have switched to using the latest and best open-source model, Qwen2.5-72B-Instruct, for our assessments and scoring. The real-time updated leaderboard can be found at `https://github.com/yzy-bupt/SVBench`.

## H  PROMPTS

Below are the prompts used for generation of QA chains within our dataset and evaluation of answers generated by all the LVLMs mentioned.

**Prompt for generating QA chains**

Task:
Please construct a chain of 5-6 consecutive open-ended QA pairs based on a series of video frames arranged in temporal order. The chain should follow the rule that, except for the first question, the content of each subsequent question must continue from the answer to the previous question. Vague questions like "What is he doing?" are permissible. If the correct answer to the previous question is not provided, the current question cannot be answered. The chain must not leapfrog to content beyond the answer to the previous question.
Generated example:
{{ "questions": [
"Did any other competitors fall?",
"How did he fall?",
"What about the other fallen competitor?",
"What does he look like?",
...
],
"answers": [
"Another rider dressed in red, white, and blue also fell.",
"He tried to avoid the first person who fell but failed.",
"The other fallen competitor ran off the track and sat down by the side of the track.", "He looks seriously injured.",
...
] }}
Required output:
- Generate the QA chain following the format of the provided example, and output the chain in JSON format. Do not produce any additional content. The QA chain should contain two key-value pairs: "questions" and "answers". "Questions" is a list of all the questions generated in the QA chain; "answers" is a list of all the answers generated in the QA chain.
- Ensure that the questions in the QA chain are open-ended and can explore multiple different aspects of the video content without focusing too much on the details. Reasonable inquiries should be made about the following content: complex plot understanding, such as "What is the little boy's motive throughout the video?"; inference of implied information, such as "How did the little boy and little girl in the video meet?"; analysis of emotional coherence, such as "At which moments in the video does the boy feel the happiest?"; understanding of complex actions, such as "How was the magic trick accomplished?"; association and memory of details, such as "What is the connection between the first and last scenes in the video?"; multi-level plot analysis, such as "What is the turning point of the plot?".
- Ensure that the questions and answers in the QA chain are strictly based on the video content itself, constructed only from the direct information in the video, avoiding any unnecessary speculation or over-association.
- Ensure that the questions in the QA chain are clear and precise, directly corresponding to specific information or events in the video, and can be answered by watching the video content without the need for a video description or inference, avoiding questions that require assumptions or reasoning.
- Ensure that the text of the questions and answers in the QA chain does not include specific time descriptions such as "which second".
- Ensure there are no references to the source of information in the QA, avoiding expressions like "from the image", "sequence of pictures", "which frame", or "which photo"; you should understand the input as a video and describe it using video footage.
- Ensure that all questions and answers are output in Chinese.
Video frame input:
{video frames}

**Prompt for generating relationships between consecutive QA chains**

Task:
You will be given two QA chains, each consisting of 5-6 consecutive open-ended QA pairs, generated for two different but consecutive segments of the same video. Your task is to identify the related QA pairs between the two QA chains.
Generated example:
{{ "questionsBefore": [
"What kind of gear are the racers wearing?",
"What kind of gear are the racers wearing?",
"What are they preparing to do?",
...
],
"answersBefore": [
"The racers are wearing motocross attire and helmets.",
"The racers are wearing motocross attire and helmets.",
"They are getting ready to start the race.",
...
],
"questionsAfter": [
"What kind of outfits are the racers wearing?",
"What kind of outfits are the racers wearing?",
"What were the racers doing at the start of the race?",
...
],
"answersAfter": [
"The racers are wearing racing suits of various colors, most equipped with safety helmets.",
"The racers are wearing racing suits of various colors, most equipped with safety helmets.",
"The racers were rushing down the hillside, ready to jump.",
...
],
"relationship":[
"Person",
"Object",
"Action",
...
] }}
Required output:
- Note that when you are looking for related QA pairs, you should search from the following six aspects: the relationship between actions, that is, when two QA pairs involve related actions; the relationship between quantities, that is, when two QA pairs involve the number of related people or objects; the relationship between persons, that is, when two QA pairs involve related persons; the relationship between objects, that is, when two QA pairs involve related objects; the relationship between events, that is, when two QA pairs involve the same or related events and activities; the relationship between environments, that is, when two QA pairs involve the same scene or changes in the scene.
- Generate results following the format of the provided example and output all results in JSON format without producing any additional content. The output should include five key-value pairs, where "questionsBefore" is a list of questions from the first QA chain within related QA pairs; "answersBefore" is a list of answers from the first QA chain within related QA pairs; "questionsAfter" is a list of questions from the second QA chain within related QA pairs;"answersAfter" is a list of answers from the second QA chain within related QA pairs; "relationship" is the list of relationships corresponding to the related QA within the two QA chains.
- Note that the number of related QA pairs in the "questionsBefore" and "questionsAfter" lists should be consistent, and the number of relationships in the "relationship" list should match the number of related QA pairs in the two QA chains.
- Note that for the two QA chains, you should output at least six related QA pairs, and each related QA pair should conform to one of the six types of relationships mentioned above.
The "relation" list should only include the following relationships:
1. "Action", when there is a relationship between actions in related QA pairs.
2. "Quantity", when there is a relationship between quantities in related QA pairs.
3. "Person", when there is a relationship between people in related QA pairs.
4. "Object", when there is a relationship between objects in related QA pairs.
5. "Event", when there is a relationship between occurrences in related QA pairs.
6. "Environment", when there is a relationship between environments in related QA pairs.
The first QA chain: {chain1}
The second QA chain: {chain2}

Prompt for generating answers of questions within our dataset by both GPT-4V and GPT-4o

Task:
You are a video comprehension expert, and you need to answer the questions posed in sequence based on the provided video image sequence. The generated answers should be concise, clear, with an emphasis on the key points, and summarized in one sentence.
Generated example: {{ "They are smiling and looking at the camera." }}
Required output:
- Ensure that the content in the answer is closely related to the topic, avoiding unnecessary expansion and redundancy to provide concise, direct, and relevant information.
- Summarize the answer clearly in one sentence, ensuring conciseness and emphasis on the key points.
- Ensure that the answer precisely targets the posed question, providing comprehensive and direct information. When answering, clearly articulate your viewpoint and ensure all content is closely related to meet the requirements of the posed question.
- Answers should be given following the provided examples, only output the answer, and do not output any text irrelevant to the answer.
- Do not provide information sources in the answer, avoid expressions like "from the image," "picture sequence," "frame number," or "picture number." You should understand the input as a video and describe it using video footage.
Video frame input: {video frames}
Posed questions: {question}

---

**Prompt for GPT4-Score**

Task Description:
You are an expert judge evaluating the accuracy of answers to question about scenes in a streaming video. For each scene, there is a specific question and its ground truth answer. Several models have provided responses to these questions. Your task is to evaluate the accuracy of each response on a scale from 0 to 10, where:
- 10: The response is completely accurate and matches the ground truth in all relevant details, providing any necessary context.
- 8-9: The response is mostly accurate but may miss minor details or context.
- 6-7: The response is somewhat accurate but lacks significant details or context.
- 4-5: The response provides some relevant information but misses key aspects of the ground truth.
- 2-3: The response has little relevance or severely misconstrues the ground truth.
- 0-1: The response is completely inaccurate or off-topic.
Additional Requirements and Considerations for the Evaluator:
1. Thoroughly Understand the Question: Ensure that you fully grasp the context and nuances of the question before evaluating the response.
2. Accurate Comparison: Compare the model's response against the ground truth answer with a high degree of precision. Pay attention to the correctness, completeness, and relevance of the information provided.
3. Objective Scoring: Assign a score on a scale from 1 to 10, focusing solely on the accuracy of the response. Do not consider style, grammar, or additional information that is unrelated to accuracy.
4. Detailed Explanation: Provide a clear and concise explanation for the score you assign. This explanation should justify your scoring by pointing out specific accurate or inaccurate details in the model's response.
5. Consistency: Apply the same criteria uniformly across all evaluations to ensure fairness and consistency in scoring.
6. Be Neutral and Unbiased: Do not let any prior knowledge, assumptions, or personal opinions affect your judgment. Only use the provided ground truth and the response when making your decision.
For the following QA, please evaluate the model's performance according to the criteria mentioned above and provide a detailed justification for each score.
Questions:
{question}
Ground Truth:
{ground_truth}
Model Responses:
{model_response}
Please provide your evaluation score and detailed comment below:
Accuracy:
Score:
Comments:

---

**Prompt for dialogue evaluation**

Task Description:
You are an evaluation expert for video multi-turn dialogue evaluation. Each video contains different timestamps that are followed by a series of QAs. Your task is to evaluate the quality of model responses to a series of open-ended questions at the same timestamp within a streaming video. You will assess the responses based on several specified dimensions:

1. Semantic Accuracy: evaluates the accuracy of the generated answers based on a holistic understanding. It considers not only the direct overlap with ground-truth answers but also the context, coherence, and overall relevance of the response to the question posed.
Scoring Guidelines: - 10 points: Completely accurate, directly reflecting the video with no apparent errors. - 7-9 points: Mostly accurate, with a few minor detail errors. - 4-6 points: Several errors, but generally conveys most of the content. - 1-3 points: Only a small part of the content is accurate or mostly incorrect. - 0 points: Completely inaccurate, unrelated to the video.

2. Contextual Coherence: examines the ability of LVLMs to maintain relevance and context across sequential questions and answers, ensuring continuity and alignment with the evolving discourse.
Scoring Guidelines: - 10 points: Highly coherent, natural transition between scenes. - 7-9 points: Mostly coherent, with minor issues in transition points. - 4-6 points: Some coherence, but loose or partially disjointed transitions. - 1-3 points: Poor coherence, most transition points unnatural. - 0 points: Completely incoherent, response appears to be unrelated or independent content.

3. Logical Consistency: evaluates the logical progression and consistency of answers, ensuring that answers do not contradict each other or previous information.
Scoring Guidelines: - 10 points: Logically consistent, no sense of incongruity. - 7-9 points: Mostly consistent, with few minor inconsistencies. - 4-6 points: Several logical issues, but the response is somewhat understandable. - 1-3 points: Logically chaotic, difficult to understand or largely unreasonable. - 0 points: Completely illogical, contradicts video content.

4. Temporal Understanding: assesses the model's proficiency in comprehending and reasoning about temporal events and sequences depicted in the video content.
Scoring Guidelines: - 10 points: The response accurately reflects the timeline and causal relationships of events. - 7-9 points: The response largely reflects the correct timeline with minor errors or omissions. - 4-6 points: The response reflects the event sequence partially but has significant time-related errors or key omissions. - 1-3 points: The response has little correct temporal understanding, with many time errors. - 0 points: The response entirely fails to reflect the correct time sequence or events process. - -1 points: The question do not involve temporal understanding

5. Informational Completeness: measures the comprehensiveness to gauge whether the model captures and conveys all relevant elements from the video to provide a thorough answer.
Scoring Guidelines: - 10 points: Fully comprehensive, covering all necessary details. - 7-9 points: Mostly comprehensive, with some missing details. - 4-6 points: Partially informative, but incomplete. - 1-3 points: Largely incomplete, containing only a few details. - 0 points: Contains no useful information.

Overall Score: is derived by aggregating the scores from each aforementioned criterion, ranked as follows:
- Scores 1-2: Irrelevant, factually incorrect, or harmful content. - Scores 3-4: Low quality, with no major errors but not meeting requirements. - Scores 5-6: Moderate quality, meets basic requirements but performs poorly in some aspects. - Scores 7-8: High quality, performs well in most dimensions. - Scores 9-10: Excellent performance, fully addressing the questions and all criteria, significantly exceeding the reference answers.

Additional Requirements and Considerations for the Evaluator: 1. Unbiased Evaluation: Ensure an unbiased assessment by focusing purely on the content and quality of the responses compared to the ground truth. 2. Consistency: Maintain consistency in scoring across different responses by adhering strictly to the detailed scoring breakdown provided. 3. Detail and Justification: Provide detailed feedback for each criterion, explaining why a particular score was assigned to help identify strengths and weaknesses in the responses. 4. Thoroughness: Avoid rushing through the evaluation. Ensure each response is carefully reviewed and scored based on all aspects of the criteria.

For the following QA chain, please evaluate the model's performance according to the criteria mentioned above and provide a detailed justification for each score.
Questions: {question}
Ground Truth: {ground_truth}
Model Responses: {model_response}
Please provide your evaluation scores and detailed comments for each criterion below:
Semantic Accuracy:
Score:
Comments:

**Prompt for streaming evaluation**

Task Description:
You are an evaluation expert for streaming video evaluation. Each video contains different timestamps that are followed by a series of QAs. Your task is to evaluate the quality of model responses to a series of open-ended questions at different timestamps within a streaming video. You will assess the responses based on several specified dimensions:
1. Semantic Accuracy: evaluates the accuracy of the generated answers based on a holistic understanding. It considers not only the direct overlap with ground-truth answers but also the context, coherence, and overall relevance of the response to the question posed.
Scoring Guidelines: - 10 points: Completely accurate, directly reflecting the video with no apparent errors. - 7-9 points: Mostly accurate, with a few minor detail errors. - 4-6 points: Several errors, but generally conveys most of the content. - 1-3 points: Only a small part of the content is accurate or mostly incorrect. - 0 points: Completely inaccurate, unrelated to the video.
2. Contextual Coherence: examines the ability of LVLMs to maintain relevance and context across sequential questions and answers, ensuring continuity and alignment with the evolving discourse.
Scoring Guidelines: - 10 points: Highly coherent, natural transition between scenes. - 7-9 points: Mostly coherent, with minor issues in transition points. - 4-6 points: Some coherence, but loose or partially disjointed transitions. - 1-3 points: Poor coherence, most transition points unnatural. - 0 points: Completely incoherent, response appears to be unrelated or independent content.
3. Logical Consistency: evaluates the logical progression and consistency of answers, ensuring that answers do not contradict each other or previous information.
Scoring Guidelines: - 10 points: Logically consistent, no sense of incongruity. - 7-9 points: Mostly consistent, with few minor inconsistencies. - 4-6 points: Several logical issues, but the response is somewhat understandable. - 1-3 points: Logically chaotic, difficult to understand or largely unreasonable. - 0 points: Completely illogical, contradicts video content.
4. Temporal Understanding: assesses the model's proficiency in comprehending and reasoning about temporal events and sequences depicted in the video content.
Scoring Guidelines: - 10 points: The response accurately reflects the timeline and causal relationships of events. - 7-9 points: The response largely reflects the correct timeline with minor errors or omissions. - 4-6 points: The response reflects the event sequence partially but has significant time-related errors or key omissions. - 1-3 points: The response has little correct temporal understanding, with many time errors. - 0 points: The response entirely fails to reflect the correct time sequence or events process. - -1 points: The question do not involve temporal understanding
5. Informational Completeness: measures the comprehensiveness to gauge whether the model captures and conveys all relevant elements from the video to provide a thorough answer.
Scoring Guidelines: - 10 points: Fully comprehensive, covering all necessary details. - 7-9 points: Mostly comprehensive, with some missing details. - 4-6 points: Partially informative, but incomplete. - 1-3 points: Largely incomplete, containing only a few details. - 0 points: Contains no useful information.
Overall Score: is derived by aggregating the scores from each aforementioned criterion, ranked as follows:
- Scores 1-2: Irrelevant, factually incorrect, or harmful content. - Scores 3-4: Low quality, with no major errors but not meeting requirements. - Scores 5-6: Moderate quality, meets basic requirements but performs poorly in some aspects. - Scores 7-8: High quality, performs well in most dimensions. - Scores 9-10: Excellent performance, fully addressing the questions and all criteria, significantly exceeding the reference answers.
Additional Requirements and Considerations for the Evaluator: 1. Unbiased Evaluation: Ensure an unbiased assessment by focusing purely on the content and quality of the responses compared to the ground truth. 2. Consistency: Maintain consistency in scoring across different responses by adhering strictly to the detailed scoring breakdown provided. 3. Detail and Justification: Provide detailed feedback for each criterion, explaining why a particular score was assigned to help identify strengths and weaknesses in the responses. 4. Thoroughness: Avoid rushing through the evaluation. Ensure each response is carefully reviewed and scored based on all aspects of the criteria.
For the following streaming QAs, please evaluate the model's performance according to the criteria mentioned above and provide a detailed justification for each score.
Questions: {question}
Ground Truth: {ground_truth}
Model Responses: {model_response}
Please provide your evaluation scores and detailed comments for each criterion below:
Semantic Accuracy:
Score:
Comments:

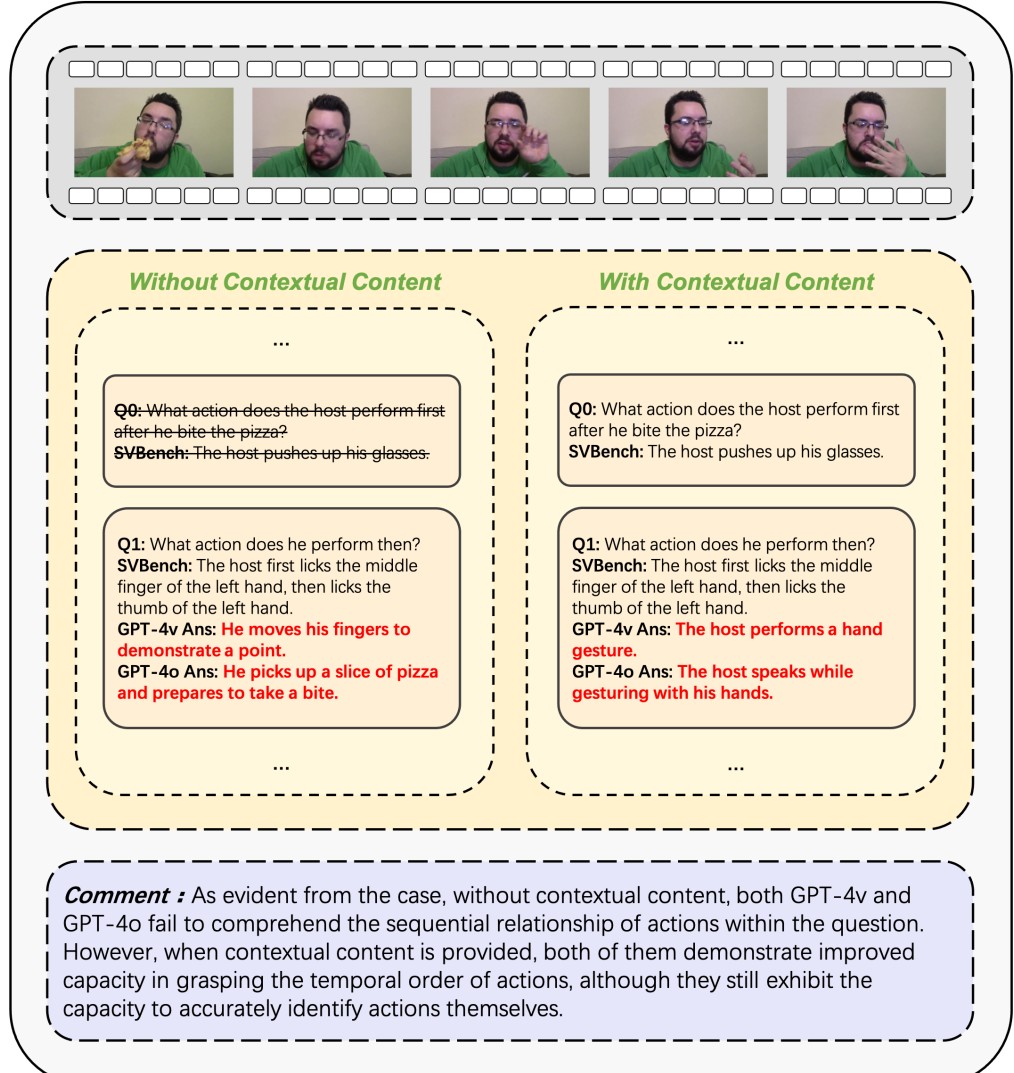

Figure 11: **Case study between a single QA pair with contextual content and the same QA pair without contextual content:** The red text highlights inaccuracies in the answer generated by both GPT-4v and GPT-4o.

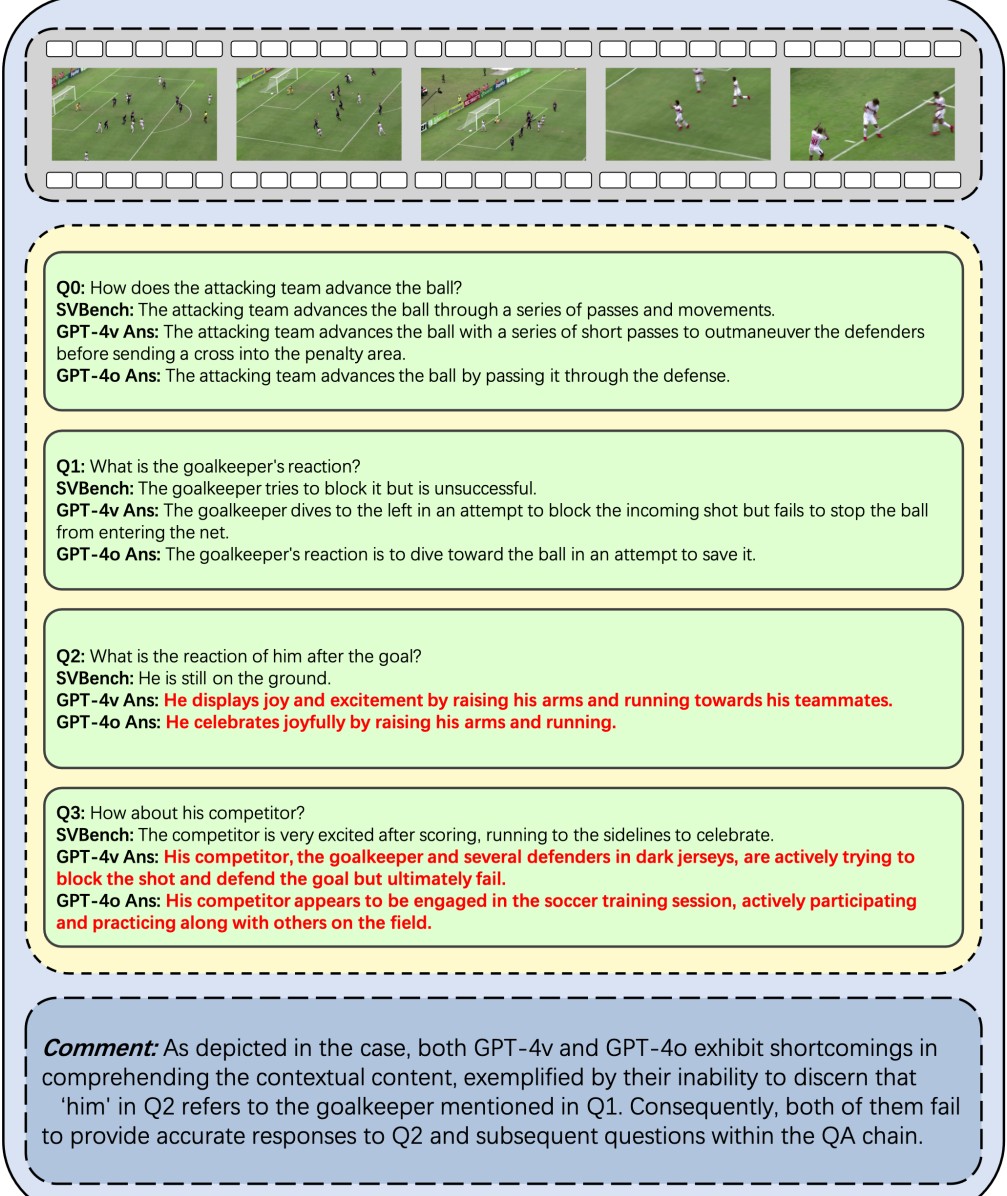

Figure 12: **Case study of a QA chain:** The red text highlights inaccuracies in answers generated by both GPT-4v and GPT-4o.

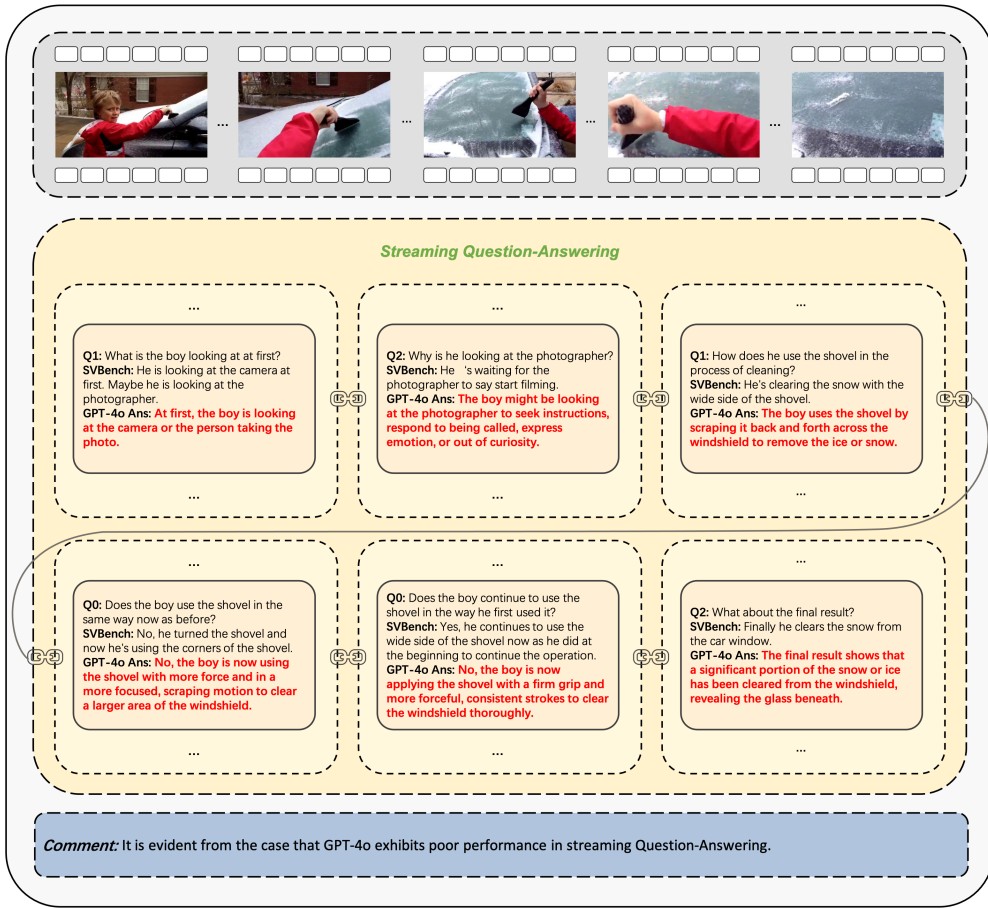

Figure 13: **Case study of Streaming Question-Answering:** The red text highlights inaccuracies in answers generated by GPT-4o.

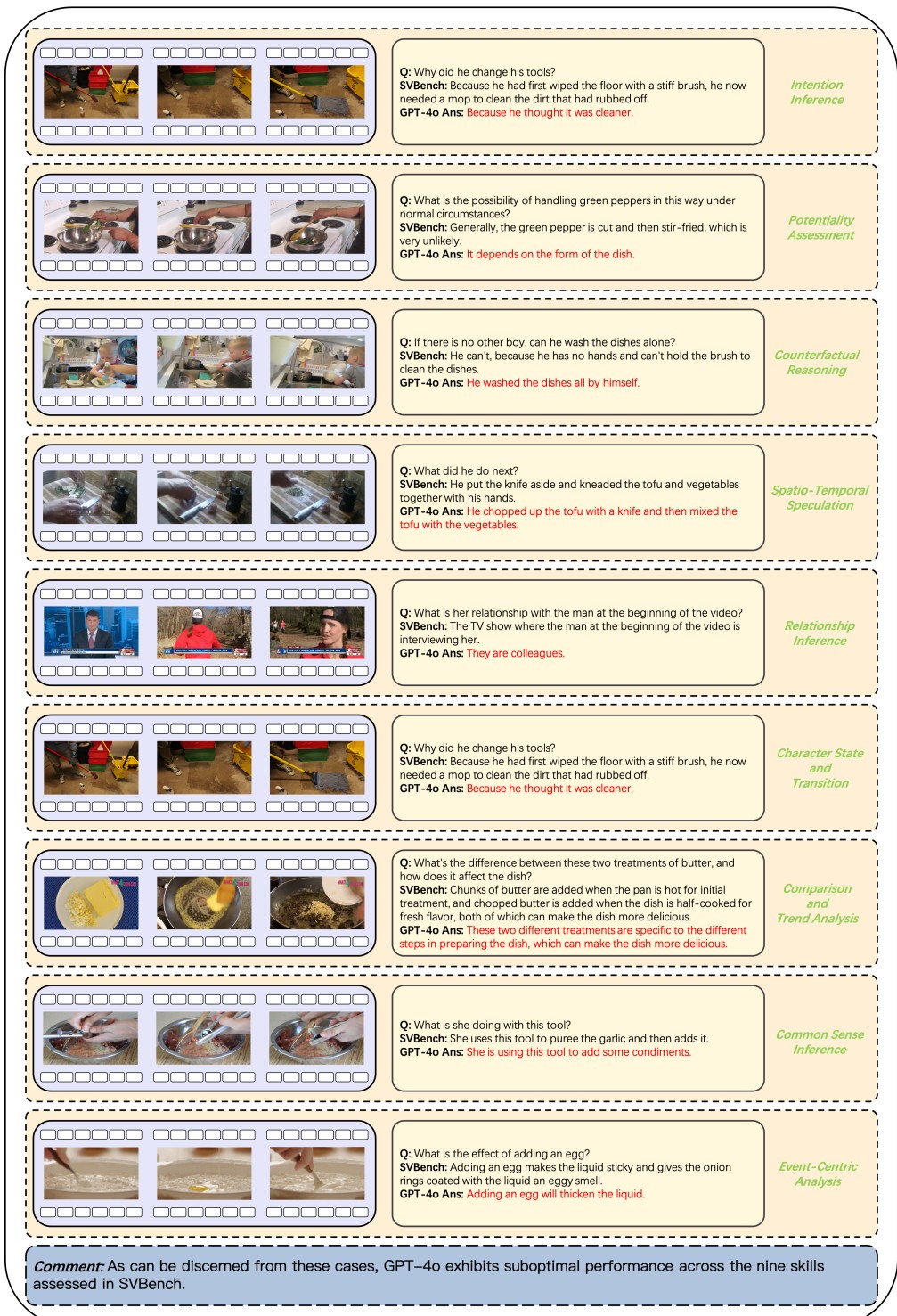

Figure 14: **Case study of 9 skills assessment:** The red text highlights inaccuracies in answers generated by GPT-4o.

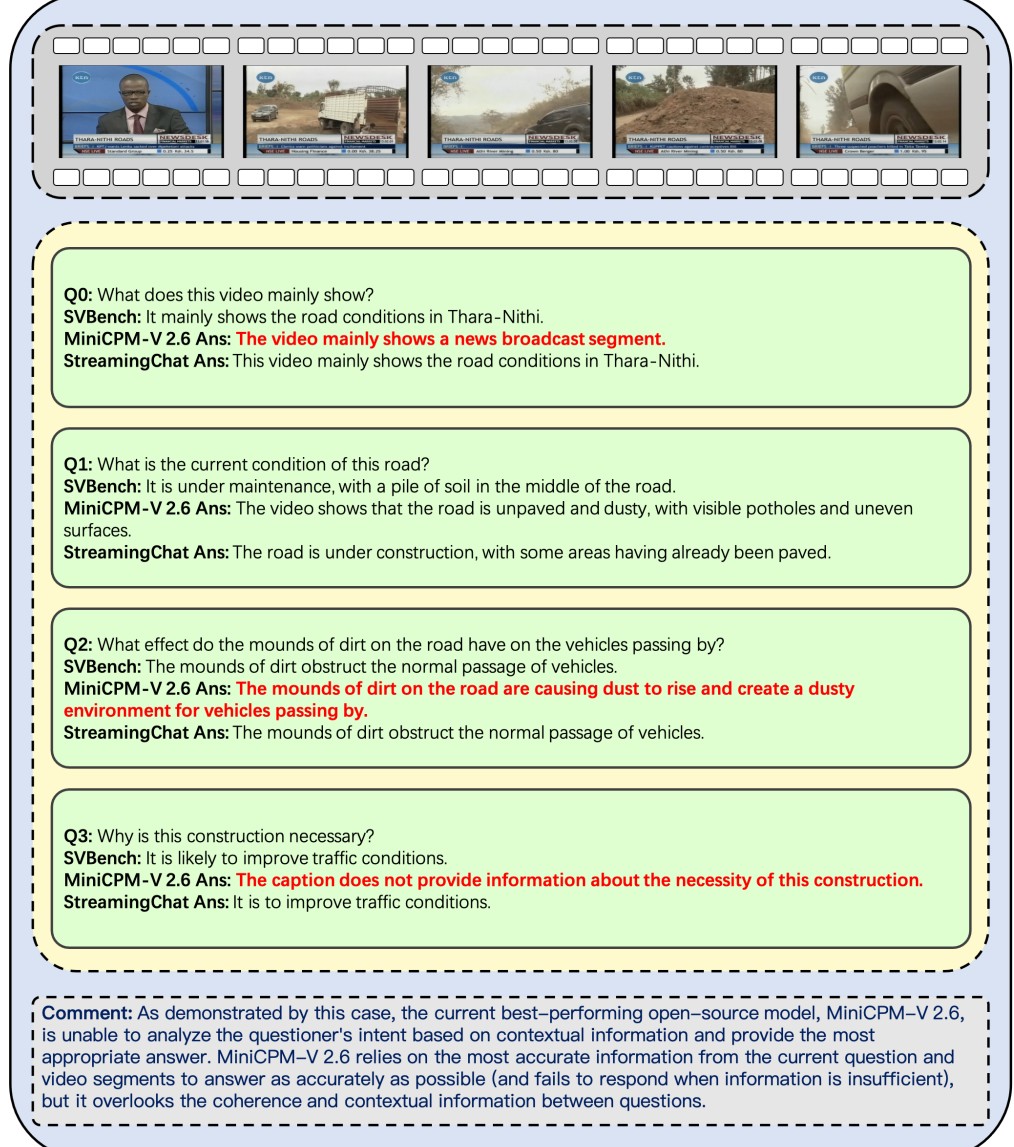

Figure 15: **Case study: Comparison of StreamingChat and the best-performing open-source LVLM on SVBench.** The red text highlights inaccuracies in answers generated by MiniCPM-V 2.6.

