# OpenReview forum: "SVBench: A Benchmark with Temporal Multi-Turn Dialogues for Streaming Video Understanding"
_ICLR.cc/2025/Conference — ICLR 2025 Spotlight_

### Official Review · Reviewer_xWou · 2024-10-28

**Soundness:** 4
**Presentation:** 3
**Contribution:** 3
**Rating:** 8
**Confidence:** 3

**Summary:**

This paper focus on evaluating LVLMs on streaming video understanding tasks. The authors claimed that existing benchmarks for video understanding merely emphasize isolated single-instance text inputs and fail to evaluate the capacity to sustain temporal reasoning throughout the entire duration of video streams. Therefore, they proposed SVBench, a comprehensive benchmark created from a semi-automated annotation pipeline to obtain QA chains that represent consecutive multi-turn dialogues. The new benchmark is closer to the real-world scenarios and the evaluation results reveal that most open-source LVLMs struggle with long-context streaming video understanding. The authors also developed StreamingChat, a novel model that could significantly outperform open-source LVLMs on SVBench and could be a good starting point to inspire future research.

**Strengths:**

1. Overall, the motivation is clear and reasonable, i.e., to develop a complete solution (benchmark + model) for streaming video understanding.
2. The paper is well-written and easy-to-follow.
3. The proposed benchmark seems to have good quality, and it's closer to real-world scenarios.

**Weaknesses:**

I only have minor concerns regarding the quality of the collected annotations and the evaluation metrics.

1. The proposed semi-automated (LLM + human) annotation pipeline is novel and reasonable. However, the authors did not provide sample data to demonstrate the high quality of the benchmark. Although they mentioned in the paper that GPT-4 was utilized to score and rank the QA chain annotations, it still cannot fully convince me that such an LLM-based check can ensure the quality of the benchmark.
2. SVBench leverages METEOR and GPT-4 Score to evaluate model outputs, which either cannot fully reveal the semantic consistency between prediction and ground truths (METEOR) or is costly to call APIs (GPT-4 Score). It would be better to explore the possibility of leveraging open-source language models (e.g., computing distances in semantic space like [1] or prompting open-source LLMs) to perform evaluation.

[1] Di et al. Grounded Question-Answering in Long Egocentric Videos. CVPR 2024.

**Questions:**

1. Can the authors provide some data samples to demonstrate the significance of the proposed benchmark?
2. Is it possible to utilize open-source models (suggested in the weakness part) to perform evaluation in SVBench? If yes, how do the results align with the existing metrics (METEOR and GPT-4 Score) and human evaluations?

---

> ### Author Response · Authors · 2024-11-22
> **Response to reviewer xWou**
>
> **For W1 and Q1: Concerns about data quality and the need for sample data to demonstrate the benchmark's significance**
>
> We understand the reviewers' concerns regarding data quality. To ensure the quality of our benchmark, we have implemented the following measures:
>
> - **Data Filtering and Cleaning:** We conducted rigorous and multiple rounds of filtering and cleaning of the raw video data, removing noise and irrelevant information to ensure high quality and richness in 3.1.
> - **Manual Annotation and Verification:** We combined automated tools with manual annotation to perform multiple rounds of verification, ensuring high-quality data in 3.1.2.
> - **Example Data:** In Appendix E of our paper, we present some example data to demonstrate the quality and diversity of our dataset. Additionally, we will soon open-source the entire dataset and models for further research.
>
> **For W2 and Q2: Concerns about METEOR and GPT-4 Score, and the suggestion to use open-source models for evaluation**
>
> We have additionally used the open-source model (InternVL2-Llama3-76B) for evaluation and compared its results with METEOR, GPT-4 Score, and human evaluations. The specific experimental results and analysis have been added to Appendix F.1 of the paper. The results indicate:
>
> 1. The score variations between human evaluations and GPT-4 are generally consistent across different models, though human scores are more discriminative, suggesting that GPT-4 scores are reasonably reliable.
> 2. InternVL2-Llama3-76B shows excessive leniency, with more than half of the models scoring above 60 using the same prompt as GPT-4, while METEOR scores are too low, lacking discrimination.
>
> Regarding the use of sentence similarity in QAEgo4D as an important evaluation metric for computing distances in semantic space, we have incorporated this reference and content into our paper in Appendix D.

---

> > ### Comment · Reviewer_xWou · 2024-11-27
> >
> > Thanks for the response from the authors. I'm holding my original rating.

---

> > > ### Author Response · Authors · 2024-12-02
> > >
> > > We greatly appreciate the time and effort you took to review our work and provide your kind feedback!

---

### Official Review · Reviewer_6qmJ · 2024-11-02

**Soundness:** 3
**Presentation:** 3
**Contribution:** 3
**Rating:** 8
**Confidence:** 4

**Summary:**

To make up for the gap of lacking attention in streaming video understanding, this paper introduces an novel designed SVBench which try to assess how well large multi-modal language models handle temporal multi-turn question-answering dialogues over streaming videos. SVBench includes 49,979 QA pairs derived from 1,353 videos and all
annotations are interconnected through temporal linkages, ensuring that the model needs to consider previous and current video segments to answer questions correctly.
The authors developed StreamingChat, which significantly improved performance by incorporating long-context reasoning abilities specific to streaming videos. They also leveraged advanced training techniques like fine-tuning with LoRA (Low-Rank Adaptation) to handle long video contexts efficiently

**Strengths:**

1. The SVBench proposed fill the gap between video benchmarks and streaming video understanding. In the real world, streaming video is a more challenging data form, so this benchmark has very important practical significance.
2. The authors proposed a semi-automatic annotation process and integrated multiple types of video data, which not only maintained the diversity of the benchmark, but also the accuracy of the annotation information and prevented hallucinations from affecting the benchmark results.
3. In response to the observed limitations of current large multi-modal langeuage models, the authors introduce StreamingChat, which significantly improves performance on SVBench.
4. The paper conducts an extensive evaluation of 14 models, comparing both open-source and closed-source models,  provides valuable insights into the current state-of-the-art models in handling streaming video understanding.

**Weaknesses:**

1. Although the authors provide an intuitive expression of the proposed SVBench in terms of video types and the diversity of annotation information through visualization results, the authors lack a description of the distribution of video lengths, which may be important for a benchmark.
2. The training method of the StreamingChat architecture proposed by the author is similar to the recently proposed streaming video understanding model Video-online. I hope the author can explain the difference between them in more detail.
3. The author's experimens are rich, but lacks measurements of the overall system's latency and inference speed like FPS, which are also very important matrics for streaming video understanding.

**Questions:**

1. According to the paper, the length of video data used by SVBench is between 5 and 30 seconds. Has the author considered the maximum video processing time of StreamingChat?
2. Can the author provide the test results of the latest streaming video understanding model Flash-VStream and Video-online on SVBench? This will further highlight the advantages of the article.
3. Can the author provide some data on the model's inference speed and resource consumption? For example, the change curve of inference speed and current consumption under different video lengths, which may be what I am more interested in.

---

> ### Author Response · Authors · 2024-11-21
> **Response to reviewer 6qmJ**
>
> Thank you so much for your positive review and insightful comments. Sorry for the wait, we've added many experiments recently into the revised version in *red*.
>
> **Weaknesses:**
>
> 1. **Lack of visualization of video length distribution:**
>    We have added a figure showing the distribution of video lengths (see Appendix A.5). The results indicate that 95.05% of the videos in the dataset are longer than 1 minute, primarily ranging from 60 to 240 seconds.
> 2. **Similarity between StreamingChat architecture and Video-online, explanation of differences:**
>    Both our proposed StreamingChat and Video-online are long-context video understanding models, utilizing interleaved multi-image training methods, which are reflected in their architecture diagrams. However, they differ significantly in model architecture, training objectives, and task definitions:
>    - **Model Architecture:** Our StreamingChat uses InternViT as the vision encoder, InternLM2 as the LLM, and an MLP projector for alignment. In contrast, Video-online uses CLIP ViT-L as the vision encoder, Llama-2-7B-Chat or Llama-3-8B-Instruct as the LLM, and an MLP projector for alignment.
>    - **Training Objectives:** StreamingChat focuses on generating answers autoregressively and accurately, while Video-online also constrains the timing of output answers (e.g., during action or event changes).
>    - **Task Definition:** StreamingChat addresses the fundamental task of streaming video understanding, answering questions in real-time as the video plays, enabling temporal multi-turn dialogue. Video-online, on the other hand, focuses on responding to a predefined question during video playback, deciding whether to answer based on action changes. Therefore, the datasets and data input formats in the architecture diagrams differ between the two models.
> 3. **Lack of evaluation on latency and FPS:**
>    We have added figures showing the inference speed and resource consumption of multiple LVLMs in relation to video length (see Appendix F.2).
>
> **Questions:**
>
> 1. **Maximum video processing time of StreamingChat:**
>    In our Statistical Analysis, we mentioned that the dataset "consists of long videos with an average length exceeding 2 minutes." The 5 to 30 seconds range refers to the duration of each scene retained during filtering to ensure video quality. In fact, 95.05% of the videos in the dataset are longer than 1 minute, primarily ranging from 60 to 240 seconds. During training, due to the context window limitation, we split temporal dialogue paths exceeding 100 frames into multiple segments (using 8 A800 GPUs with 80G each). At 1 FPS, the maximum duration for each data segment is 100 seconds.
> 2. **Testing the latest Flash-VStream and Video-online on SVBench:**
>    We have added the results of Flash-VStream in Appendix F.3 (to be incorporated into the main text later). However, as mentioned in W2, Video-online is not suitable for our defined streaming video understanding task, as it focuses on responding to predefined questions during video playback rather than answering questions in real-time (hindering multi-turn QAs). We will consider including it in future work.
> 3. **Comparison of inference speed and resource consumption:**
>    We have added figures showing the inference speed and resource consumption of multiple LVLMs in relation to video length (see Appendix F.2). The inference time is generally positively correlated with the video length. For resource consumption, some models show a plateau as the input length increases, due to the saturation of input frames and the memory reaching its preset limit.

---

> ### Author Response · Authors · 2024-12-02
>
> Thank you so much for your positive review and insightful comments. We have carefully considered your comments and provided detailed responses. We look forward to your feedback.

---

> > ### Comment · Reviewer_6qmJ · 2024-12-02
> >
> > Thank you for your response. Most of my concerns have been addressed, and I will increase my rating to 8.

---

> > > ### Author Response · Authors · 2024-12-02
> > >
> > > We greatly appreciate your time and effort in reviewing our work. Thank you very much for your swift response and for increasing your score!

---

### Official Review · Reviewer_k44W · 2024-11-03

**Soundness:** 3
**Presentation:** 4
**Contribution:** 3
**Rating:** 8
**Confidence:** 4

**Summary:**

In this paper, the authors introduce SVBench, a novel benchmark designed to evaluate the capabilities of LVLMs in long-context streaming video understanding. SVBench features temporal multi-turn question-answering chains comprising 49,979 QA pairs across 1,353 streaming videos. The study reveals that while the closed-source GPT-4o model outperforms others, most open-source LVLMs struggle with long-context streaming video understanding.

The authors also develop StreamingChat, a model that outperforms existing open-source LVLMs on SVBench and achieves comparable performance on diverse vision-language benchmarks.

**Strengths:**

S1: The paper introduces SVBench, a benchmark explicitly designed for evaluating LVLMs in long-context streaming video understanding. I believe this fills a notable gap in existing benchmarks, which typically focus on isolated text inputs rather than sustained temporal reasoning across video streams. The comparison between the benchmarks also shows the advantages of SVBench.

S2: The QA pairs in this work are annotated semi-automatically, making it a large-scale dataset with high-quality annotations.

S3: The experiments and evaluation provide insights into open-source models' struggles with long-context video understanding.

S4: The paper is well-organized, with clear explanations of SVBench, the QA chain structure, and the temporal linkages created for sustained reasoning tasks.

**Weaknesses:**

W1: The paper does not include a comparison with human performance. Incorporating such a comparison would provide valuable insights into the gap between current models and human capabilities in long-context streaming video understanding.

W2: The paper does not analyze the impact of language model size on performance. Considering that models like InternVL2 have versions with 1B, 2B, 4B, 8B, 26B, 40B, and 72B parameters, and Video-LLaMA2 also have 72B versions, expanding experiments to include these variations and providing more detailed analysis would enhance the work. Additionally, exploring the number of frames the model can process would offer valuable insights. In addition to model size and the input length, analyzing the amount and diversity of training data used for each model would provide a more comprehensive understanding of their performance in long-context streaming video understanding. Training data quality and quantity are crucial factors influencing model capabilities.

W3: The evaluation overlooks several important models capable of long-video understanding, such as LLaVA-OneVision [1], Qwen2-VL [2], LongVILA [3], Long-LLaVA [4], and Oryx [5]. Including these models would enhance the comparison, providing a more comprehensive view of current capabilities in this area.

[1] https://github.com/LLaVA-VL/LLaVA-NeXT

[2] https://github.com/QwenLM/Qwen2-VL

[3] https://github.com/NVlabs/VILA/blob/main/LongVILA.md

[4] https://github.com/FreedomIntelligence/LongLLaVA

[5] https://github.com/Oryx-mllm/Oryx

**Questions:**

Q1: The videos of the benchmark come from some publicly available datasets; how do you ensure that the models used for evaluation have not encountered this data during their training?

Q2: How do QA chains address the issue of information sparsity in videos? For lengthy videos with few key events, how does SVBench construct meaningful QA chains? Is there a mechanism to handle or generate question chains that effectively deal with scenarios where information is sparse or low-density?

Q3: In your benchmark, do the videos retain their audio tracks? Multimodal information significantly aids in understanding videos. If audio is included, does it genuinely assist in answering the questions?

---

> ### Author Response · Authors · 2024-11-22
> **Response to reviewer k44W**
>
> Thank you so much for your positive review and insightful comments. Sorry for the wait, we've added many experiments recently into the revised version in red.
>
> **Weaknesses:**
>
> 1. Lack of comparison with human performance
>
>    We have included a comparison with human performance, as shown in Table 6 of Appendix F.3. This comparison provides insights into the gap between current models and human capabilities in long-context streaming video understanding. The results indicate that human performance significantly outperforms various open-source and closed-source models across all metrics in both evaluation settings (Dialogue Evaluation and Streaming Evaluation). Additionally, humans excel in Temporal Understanding (TU) but perform relatively weaker in Informational Completeness (IC). We also included LVLMs such as Flash-VStream-7B, Qwen2-VL-7B, and LLaVA-NeXT-Video-7B-DPO.
>
> 2. Lack of analysis on the impact of model size on performance
>
>    To ensure a fair comparison, we selected base models with 7B or 8B parameters. Our benchmark is an ongoing project, and we plan to update it with different sizes of mainstream Video-LLMs, maintaining a real-time updated leaderboard. Additionally, we conducted further experiments comparing the performance of models with different LLM sizes, specifically InternVL2-2B, 4B, 8B, 26B, 40B, and 76B, on SVBench. The results indicate that model size significantly impacts performance, with larger models demonstrating better accuracy. However, the computational resource requirements also increase correspondingly. The specific experimental results and analysis have been added to Appendix F.4 of the paper.
>
> 3. Lack of comparison with several important models
>
>    Thank you for pointing this out. We have now included the evaluation results for LLaVA-NeXT and QwenVL2 in Appendix F.3 of the paper. We will also add the evaluation results for the remaining models to Table 6 as soon as possible.
>
> **Questions:**
>
> 1. The benchmark videos come from publicly available datasets; how do you ensure that the models used for evaluation have not encountered this data during their training?
>
>    Our dataset specifically focuses on streaming understanding tasks. We have constructed a novel and reasonable annotation pipeline to create unique QA chains, ensuring that the data has not been used for training other models. However, we acknowledge that it is challenging to guarantee that no model has been trained on these videos, as high-quality datasets are often publicly accessible. We have employed a complex filtering process to select videos from multiple large-scale datasets (e.g., YT-Temporal-1B), ensuring that the evaluation models do not cover all the videos we use. In the future, we plan to crawl the latest videos or shoot our own to ensure this.
>
> 2. How do QA chains address the issue of information sparsity in videos?
>
>    We have removed sparse or low-density videos during the dataset construction process. Only videos with an average scene duration between 5 and 30 seconds are chosen, ensuring fluidity and rhythm essential for effective streaming data analysis. This ensures that the QA chains can describe temporal changes meaningfully, as constructing meaningful QA chains for long videos without key events is challenging.
>
> 3. Do the videos in your benchmark retain their audio tracks? Does audio genuinely assist in answering the questions?
>
>    Our work aims to evaluate Large Vision-Language Models (LVLMs), so we selected baselines that do not consider audio to ensure a fair comparison. Since most of the models we selected cannot incorporate audio information, for models that can, such as VideoLLaMA2, we chose the VideoLLaMA2-7B-16F model instead of the VideoLLaMA2.1-7B-AV model, which can include audio information. Although we have not yet conducted experiments with audio, existing research suggests that retaining audio information can improve overall model performance, particularly in complex tasks requiring the integration of multiple information sources. For instance, the results in Figure 8 of [1] indicate that incorporating audio information generally enhances evaluation results on video QA benchmarks.
>
> [1] Shu F, Zhang L, Jiang H, et al. Audio-visual LLM for video understanding[J]. arXiv preprint arXiv:2312.06720, 2023.

---

> ### Comment · Reviewer_k44W · 2024-12-01
>
> Thank you so much for your responses. Your experiments are now much more comprehensive, and all my concerns have been addressed. Therefore, I'd like to maintain my current rating.
>
> I'm also curious about your future research plans based on your benchmark. I’m also curious about your future research plans based on this benchmark. Do you intend to extend it to encompass additional streaming video understanding tasks, such as streaming grounding or streaming dense captioning?

---

> > ### Author Response · Authors · 2024-12-02
> > **Response to reviewer k44W**
> >
> > Thank you for your time and positive feedback.
> >
> > Those are excellent ideas. Yes, we are indeed extending our research to encompass additional tasks related to streaming video understanding. We have designed a new data annotation pipeline and benchmark for streaming captioning, storytelling, and narration. Our team is actively working on streaming captioning models. Please stay tuned!

---

### Official Review · Reviewer_YPvF · 2024-11-04

**Soundness:** 3
**Presentation:** 3
**Contribution:** 3
**Rating:** 6
**Confidence:** 3

**Summary:**

This paper introduces SVBench, a benchmark designed to evaluate Large Vision-Language Models (LVLMs) on streaming video understanding. A single data point in SVBench consists of a video segment with its corresponding temporal dialogue chain. Each chain contains 4-5 question-answer pairs that are contextually connected, meaning each subsequent question builds upon previous answers. Additionally, each chain has "temporal linkages" to chains from adjacent video segments based on common elements (people, events, objects).

The authors employed a semi-automated annotation pipeline where GPT-4 generates QA pairs for each video segment, identifies related QA pairs based on six relationship types, followed by human annotators who modify the result. The evaluation framework uses two modes - dialogue evaluation (testing multi-turn QA within the same video segment) and streaming evaluation (testing understanding across temporally linked segments with an 80% chance of jumping to questions linked with temporal linkages) - while scoring responses across five dimensions using LLMs.

The authors also introduced StreamingChat, a LVLM built on InternVL2 that uses an InternViT vision encoder, MLP projector, and InternLM2 language model. Regarding the results, GPT-4o leads with the highest scores (66.29% in dialogue and 58.17% in streaming evaluation) while among open-source models, StreamingChat performs best (59.41% in dialogue and 53.90% in streaming), with all models scoring below 60%, indicating significant room for improvement in streaming video comprehension.

**Strengths:**

- Novel Technical Contribution: The paper introduces a semi-automated pipeline that combines LLM-assisted generation with human verification to create temporal multi-turn QA chains, representing a methodologically sound approach to dataset creation.

- Comprehensive Empirical Validation: The evaluation spans 14 different models (both open and closed-source), uses multiple metrics (METEOR, GPT4-Score, etc.), and includes detailed ablation studies comparing single-instance vs. multi-turn QA performance.

- The temporal linkage: The temporal linkage concept between QA chains is interesting and addresses a real gap in video understanding benchmark by forcing models to maintain context across time segments, mimicking real-world streaming scenarios.

**Weaknesses:**

The paper's heavy reliance on LLMs for evaluation is a significant methodological concern. While authors use GPT-4 to assess models’ performance across multiple dimensions (semantic accuracy, contextual coherence, etc.), there's no validation of whether these automated scores align with human judgments. The fact that LLMs are being used to both generate the annotations and evaluate the results creates a circular dependency that could mask real limitations or biases in the evaluation process. Without human validation studies, it's difficult to trust the reported performance gains or confidently compare different models using this benchmark.

**Questions:**

- How reliable are LLMs evaluations?
- Did you try different LLMs to evaluate the models?
- How did evaluation depend on LLMs hyperparameters like temperature?
- Does it change if run several times?
- How does it change over different prompts?

---

> ### Author Response · Authors · 2024-11-21
> **Response to reviewer YPvF**
>
> Thank you so much for your positive review and insightful comments. We appreciate your feedback and suggestions, which we have incorporated into the revised version in *red*.
>
> **Weaknesses:**
> The evaluation heavily relies on LLMs. Using GPT-4 for both generation and validation creates a circular dependency.
> **Response:**
> We have addressed this by incorporating human involvement in the generation process and human evaluation to ensure consistency with LLM assessments. We have added human evaluations of various LVLMs results (see Appendix F.1). To validate the consistency between human scores and those from open-source and closed-source models, we employed human evaluation (10 professional annotators annotating 200 videos over a week), open-source model evaluation (InternVL2-Llama3-76B), closed-source evaluation model (GPT-4), and traditional evaluation metrics (METEOR). We then plotted a score comparison on Multi-turn QA Evaluation. The results indicate: (1) The score variations between Human and GPT-4 are generally consistent across different models, though human scores are more discriminative, suggesting that GPT-4 scores are reasonably reliable; (2) InternVL2-Llama3-76B shows excessive leniency, with more than half of the models scoring above 60 using the same prompt as GPT-4, while METEOR scores are too low, lacking discrimination.
>
> Regarding the circular dependency issue, we mitigated the reliance on LLMs by incorporating two stages of manual annotations during the construction of annotations. Additionally, the inclusion of open-source and human evaluations enriched the evaluation results, further reducing the dependency on GPT-4 for evaluation.
>
> **Questions:**
>
> 1. **How reliable are LLM evaluations?**
>    In our evaluation results (see Appendix F.1), the score variations between GPT-4 and Human are generally consistent across different models, suggesting that GPT-4 scores are reasonably reliable.
> 2. **Did you try different LLMs to evaluate the models?**
>    Yes, we used an open-source model (InternVL2-Llama3-76B) with the same prompt as GPT-4 for evaluation. The results showed excessive leniency, leading to higher average scores for all models.
> 3. **How did evaluation depend on LLMs hyperparameters like temperature?**
>    In fact, under the appropriate setting of averaging multiple runs, adjusting the temperature had little impact on the results. We fixed GPT-4's temperature at 0.7.
> 4. **Does it change if run several times?**
>    For LLM evaluations, we averaged the results over 5 runs. For human evaluations, we used the mean scores from ten annotators.
> 5. **How does it change over different prompts?**
>    In our experiments, more detailed prompts resulted in greater score discrimination. Initially, our prompts did not specify the scoring details for each score range, leading to Multi-turn QA Evaluation scores clustering between 40 and 50. After we detailed the specific criteria for each score range (see Appendix G), the score discrimination improved slightly.

---

> > ### Author Response · Authors · 2024-12-02
> >
> > Thank you for taking the time and effort to review our submission. We have carefully considered your comments and provided detailed responses. We look forward to your feedback.

---

### Meta-Review · Area_Chair_KrRu · 2024-12-18

**Metareview:**

This paper examines the performance of VLMs on streaming video understanding tasks. The authors argue that current benchmarks for video understanding fail to assess the ability to maintain temporal reasoning over the entire duration of video streams. To address this, they introduced SVBench, a benchmark developed using a semi-automated annotation process to generate QA chains representing sequential multi-turn dialogues. The reviewers highlighted the technical contribution, the comprehensive evaluation and the clear writing.
It further addresses a key gap in the literature and is a valuable contribution to the community and the AC recommends acceptance.

**Additional Comments On Reviewer Discussion:**

Initial concerns were raised about test-set leakage, further quantitative comparisons (especially wrt to larger model sizes) and model differentiation compared to previous works but were resolved during the rebuttal phase leading to increased scores.

---

### Decision · Program_Chairs · 2025-01-22

Accept (Spotlight)